# Generalized Linear Bandits with Local Differential Privacy

**Yuxuan Han**[1*]
yhanat@connect.ust.hk

**Zhipeng Liang**[2*]
zliangao@connect.ust.hk

**Yang Wang**[1,2]
yangwang@ust.hk

**Jiheng Zhang**[1,2]
jiheng@ust.hk

Department of Mathematics[1]
Department of Industrial Engineering and Decision Analytics[2]
The Hong Kong University of Science and Technology

## Abstract

Contextual bandit algorithms are useful in personalized online decision-making. However, many applications such as personalized medicine and online advertising require the utilization of individual-specific information for effective learning, while user's data should remain private from the server due to privacy concerns. This motivates the introduction of local differential privacy (LDP), a stringent notion in privacy, to contextual bandits. In this paper, we design LDP algorithms for stochastic generalized linear bandits to achieve the same regret bound as in non-privacy settings. Our main idea is to develop a stochastic gradient-based estimator and update mechanism to ensure LDP. We then exploit the flexibility of stochastic gradient descent (SGD), whose theoretical guarantee for bandit problems is rarely explored, in dealing with generalized linear bandits. We also develop an estimator and update mechanism based on Ordinary Least Square (OLS) for linear bandits. Finally, we conduct experiments with both simulation and real-world datasets to demonstrate the consistently superb performance of our algorithms under LDP constraints with reasonably small parameters $(\varepsilon, \delta)$ to ensure strong privacy protection.

## 1 Introduction

Contextual bandit algorithms have received extensive attention for their efficacy for online decision making in many applications such as recommendation system, clinic trials, and online advertisement [7, 35, 24]. Despite their success in many applications, intensive utilization of user-specific information, especially in privacy-sensitive domains such as clinical trials and e-commerce promotions, raises concerns about data privacy protection. Differential privacy, as a provable protection against identification from attackers [18, 19], has been put forth as a competitive candidate for a formal definition of privacy and has received considerable attention from both academic research [33, 17, 43, 36, 8] and industry adoption [20, 12, 37]. While increasing attention has been paid to bandit algorithms with *joint differential privacy* [34, 9], we introduce in this paper a more stringent notion, *local differential privacy* (LDP), in which users even distrust the server collecting the data, to contextual bandits.

---

*Equal contributions.

35th Conference on Neural Information Processing Systems (NeurIPS 2021).

In contextual bandit, at each time round $t$ with individual-specific context $X_t$, the decision maker can take an action $a_t$ from a finite set (arms) to receive a reward randomly generated from the distribution depending on the context $X_t$ and the chosen arm through its parameter $\theta^\star_{a_t}$ which is not unknown to the decision maker. We use the standard notion of expected regret to measure the difference between expected rewards obtained by the action $a_t$ and the best achievable expected reward in this round. While several papers consider the adversarial setting (i.e., $X_t$ can be arbitrary determined in each round), this paper considers the stochastic contextual case where $X_t$ is generated i.i.d. from a distribution $P_X$. The goal is to maximize the rewards accumulated over the time horizon. An algorithm achieves LDP guarantee if every user involved in this algorithm is guaranteed that anyone else can only access her context (and related information such as the arm chosen and the reward) with limited advantage over a random guess. Recently there is an emerging steam of works combining LDP and bandit. [6, 29, 10] consider the LDP contextual-free bandit and design algorithms to achieve the same regret as in the non-privacy setting. For contextual bandits, [44] considers the adversarial setting. Despite their pioneering work, their regret bounds $O(T^{3/4})$ leave a gap from the corresponding non-privacy results $O(T^{1/2})$, which is conjectured to be inevitable. A natural question arises: can we close this gap for stochastic contextual bandits? In this paper, we design several algorithms and show that they can achieve the same regret rate in terms of $T$ as in the non-private settings.

If we don't assume any structure on the arms' parameters, the above formulation is referred to as multi-parameter contextual bandits. If we impose structural assumptions such as all arms share the same parameter (see Section 2.2 for details), then the formulation is referred to as single-parameter contextual bandits. Although multi-parameter and single-parameter settings can be shown to be equivalent, they need independent analysis and design of algorithms because of their distinct properties based on different modeling assumptions (e.g., [27]). In this paper, we consider the privacy guarantee in both settings. In fact, multi-parameter setting is more difficult since we need to estimate the parameters for all $K$ arms with sufficient accuracy to make good decisions. However, privacy protection also requires protecting the information about which arm is pulled in each round. Such a requirement hinders the identification of optimal arm and may incur considerable regret in the decision process. A proper balance between privacy protection and estimation accuracy is the key to design algorithms with desired performance guarantee in this setting.

| Result | Regret | Context | Parameter | $\beta$-Margin |
|---|---|---|---|---|
| Theorem 10 [44] | $\tilde{O}(T^{3/4}/\varepsilon)$ | Adversary | Both | No Margin |
| Theorem 3.1 | $\tilde{O}(T^{1/2}/\varepsilon)$ | Stochastic | Single | No Margin |
| Theorem 3.3 | $O(\log T/\varepsilon^2)$ | Stochastic | Single | $\beta = 1$ |
| Theorem 3.3 | $\tilde{O}(T^{\frac{1-\beta}{2}}/\varepsilon^{1+\beta})$ | Stochastic | Single | $0 \leq \beta < 1$ |
| Theorem 4.1 | $O((\log T/\varepsilon)^2)$ | Stochastic | Multiple | $\beta = 1$ |
| Theorem 4.1 | $\tilde{O}(T^{\frac{1-\beta}{2}}/\varepsilon^{1+\beta})$ | Stochastic | Multiple | $0 < \beta < 1$ |

Table 1: Summary of our main results in $(\varepsilon, \delta)$-LDP, where $\tilde{O}(\cdot)$ omits poly-logarithmic factors.

**Contributions.** We organize our results for various settings in Table 1. Our main contributions can be summarized as follows:

1. We develop a framework for implementing LDP algorithms by integrating greedy algorithms with a private OLS estimator for linear bandits and a private SGD estimator for generalized linear bandits. We prove that our algorithms achieve regret bound matching the corresponding non-privacy results.

2. In the multi-parameter setting, to ensure the privacy of the arm pulled in each round, we design a novel LDP strategy by simultaneously updating all the arms with synthetic information instead of releasing the pulled arm. By conducting such synthetic updates for unselected arms, we protect the information of the pulled arm from being identified by the server or other users. This is at the cost of corrupting the estimation of the un-selected arms. To deal with this issue, we design an elimination method that is only based on data collected during a short warm up period. We show that such a mechanism can be combined with the OLS and SGD estimators to achieve the desired performance guarantees.

3. We introduce the SGD estimator to bandit algorithms to tackle generalized linear reward structure. To the best of our knowledge, few papers have ever considered SGD-based bandit algorithms. Theoretical regret bounds are established in [13] by combining SGD and Thompson Sampling, while most of the others are limited to empirical studies [7, 32]. We establish such theoretical regret bounds for SGD-based bandit algorithms. Our private SGD estimator for bandits is highly computationally efficient, and more importantly, greatly simplifies the data processing mechanism for LDP guarantee.

## 2 Preliminaries

**Notations.** *We start by fixing some notations that will be used throughout this paper. For a positive integer $n$, $[n]$ denotes the set $\{1, \cdots, n\}$. $|A|$ denotes the cardinality of the set $A$. $\|\cdot\|_2$ is Euclidean norm. $W(i, j)$ denotes the element in the $i$-th row and $j$-th column of matrix $W$. We write $W > 0$ if the matrix $W$ is symmetric and positive definite. We denote $I_d$ as the $d$-dimensional identity matrix. Let $\otimes$ denote the Kronecker product. Let $B_r^d$ denote the $d$-dimensional ball with radius $r$ and $S_r^{d-1}$ denotes the $(d-1)$-dimensional sphere for the ball. Given a set $A$, Unif$(A)$ denote the uniform distribution over $A$. For a tuple $(Z_{i,j})_{i \le N, j \le M}$ and $1 \le k_1 < k_2 \le M$, we denote $Z_{i,k_1:k_2} = (Z_{i,k_1}, \cdots, Z_{i,k_2})$. We adopt the standard asymptotic notations: for two non-negative sequences $\{a_n\}$ and $\{b_n\}$, $\{a_n\} = O(\{b_n\})$ iff $\limsup_{n \to \infty} a_n/b_n < \infty$, $a_n = \Omega(b_n)$ iff $b_n = O(a_n)$, $a_n = \Theta(b_n)$ iff $a_n = O(b_n)$ and $b_n = O(a_n)$. We also write $\tilde{O}(\cdot)$, $\tilde{\Omega}(\cdot)$ and $\tilde{\Theta}(\cdot)$ to denote the respective meanings within multiplicative logarithmic factors in $n$.*

### 2.1 Local Differential Privacy

**Definition 2.1** (Local differential privacy). *We say a (randomized) mechanism $M : \mathcal{X} \to \mathcal{Z}$ is $(\varepsilon, \delta)$-LDP, if for every $x \ne x' \in \mathcal{X}$ and any measurable set $C \subset \mathcal{Z}$ we have*

$$P(M(x) \in C) \le e^\varepsilon P(M(x') \in C) + \delta.$$

*When $\delta = 0$, we simply denote $\varepsilon$-LDP.*

We now present some tools that will be useful for our analysis.

**Lemma 2.1** (Gaussian Mechanism [16, 19]). *For any $f : \mathcal{X} \to \mathbb{R}^n$, let $\sigma_{\varepsilon, \delta} = \frac{1}{\varepsilon} \sup_{x, x' \in \mathcal{X}} \|f(x) - f(x')\|_2 \sqrt{2 \ln(1.25/\delta)}$. The Gaussian mechanism, which adds random noise independently drawn from distribution $\mathcal{N}(0, \sigma_{\varepsilon, \delta}^2 I_n)$ to each output of $f$, ensures $(\varepsilon, \delta)$-LDP.*

Besides the Gaussian mechanism, we also use the following privacy mechanism for bounded vectors.

**Lemma 2.2** (Privacy Mechanism for $l_2$-ball [14]). *For any $R > 0$, let $r_{\varepsilon, d} = R \frac{\sqrt{\pi}}{2} \frac{e^\varepsilon + 1}{e^\varepsilon - 1} \frac{d \Gamma(\frac{d+1}{2})}{\Gamma(\frac{d}{2} + 1)}$ where $\Gamma$ is the Gamma function. For any $x \in B_R^d$, consider the mechanism $\Psi_{\varepsilon, R} : B_R^d \to S_{r_{\varepsilon, d}}^{d-1}$ of generating $Z_x$ as the follows. First, generate a random vector $\tilde{X} = (2b - 1)x$ where $b$ is a Bernoulli random variable with success probability $\frac{1}{2} + \frac{\|x\|_2}{2R}$. Next, generate random vector $Z_x$ via*

$$Z_x \sim \begin{cases} \text{Unif}\{z \in \mathbb{R}^d : z^T \tilde{X} > 0, \|z\|_2 = r_{\varepsilon, d}\} \text{ with probability } e^\varepsilon/(1 + e^\varepsilon), \\ \text{Unif}\{z \in \mathbb{R}^d : z^T \tilde{X} \le 0, \|z\|_2 = r_{\varepsilon, d}\} \text{ with probability } 1/(1 + e^\varepsilon). \end{cases}$$

*Then $\Psi_{\varepsilon, R}$ is $\varepsilon$-LDP and $\mathbb{E}[\Psi_{\varepsilon, R}(x)] = x$.*

**Lemma 2.3** (Post-Processing property [19]). *If $M : \mathcal{X} \to \mathcal{Y}$ is $(\varepsilon, \delta)$-LDP and $f : \mathcal{Y} \to \mathcal{Z}$ is a fixed map, then $f \circ M : \mathcal{X} \to \mathcal{Z}$ is $(\varepsilon, \delta)$-LDP.*

**Lemma 2.4** (Composition property [19]). *If $M_1 : \mathcal{X} \to \mathcal{Z}_1$ is $(\varepsilon_1, \delta_1)$-LDP and $M_2 : \mathcal{X} \to \mathcal{Z}_2$ is $(\varepsilon_2, \delta_2)$-LDP, then $M = (M_1, M_2) : \mathcal{X} \to \mathcal{Z}_1 \times \mathcal{Z}_2$ is $(\varepsilon_1 + \varepsilon_2, \delta_1 + \delta_2)$-LDP.*

### 2.2 Local Differential Privacy in Bandit

We consider contextual bandits with LDP guarantee in the context of the user-server communication protocol described in Figure 1. The user in round $t$ with context $X_t \in \mathbb{R}^d$ receives (processed) historical information $S_{t-1}$ from the server, and chooses an action $a_t \in [K]$ to obtain a random reward $r_t = v(X_t, a_t) + \epsilon_t$ . Define $\mathcal{F}_t$ as the filtration of all historical information up to time

$t$, i.e., $\mathcal{F}_t = \sigma(X_1, \cdots, X_t, \epsilon_1, \cdots, \epsilon_{t-1})$, and we require $\epsilon_t$ is bounded and $\mathbb{E}[\epsilon_t|\mathcal{F}_t] = 0$. Then the user processes the tuple $(X_t, r_t)$ by some mechanism $\psi$ with LDP guarantee and send the processed information $Z_t = \psi(X_t, r_t)$ to the server. After receiving $Z_t$, the server updates the historical information $S_t$ to get $S_{t+1}$. We consider the generalized linear bandits by allowing $v(X_t, a_t) = \mu(X_t^T \theta_{a_t}^\star)$, where $\mu : \mathbb{R} \to \mathbb{R}$ is a link function and $\theta_i^\star \in \mathbb{R}^d$ is the underlying parameter of the $i$-th arm. For a fix time t, we denote $a_t^* = \arg\max_{i \in [K]} \mu(X_t^T \theta_i^\star)$. The regret over time horizon $T$ is $\mathrm{Reg}(T) = \sum_{t=1}^T \left( \mu(X_t^T \theta_{a_t^*}^\star) - \mu(X_t^T \theta_{a_t}^\star) \right)$. If we don't assume any structure on $\{\theta_i^\star\}_{i \in [K]}$, we refer it as the multi-parameter setting. We also consider $d$-dimensional single-param setting by assuming $\theta_i^\star = e_i \otimes \theta^\star$ for some $\theta^\star \in \mathbb{R}^d$ where $\{e_i\}_{i \in [K]}$ is canonical basis of $\mathbb{R}^K$. In this case, $x_{t,i} \in \mathbb{R}^d$ is the $i$-th segment of $X_t \in \mathbb{R}^{dK}$ and $X_t^T \theta_i^\star = x_{t,i}^T \theta^\star$, so choosing arm $i$ becomes choosing the $i$-th segment $x_{t,i}$ of the context.

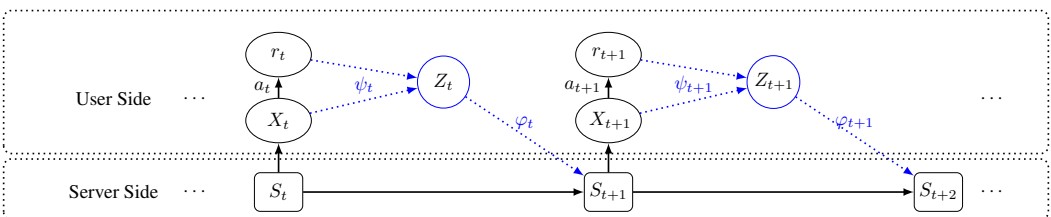

Figure 1: User-server communication protocol

In the rest of paper, we always assume that $\|\theta_i^\star\|_2 \le 1, \forall i \in [K]$, the reward is bounded by $c_r$ and the Euclidean norm of the context is bounded by $C_B$, our analysis can be easily generalized to the case where $\epsilon_t$ and the context follow sub-gaussian distributions. In fact, when the context and noise are subgaussian, it is guaranteed by the subgaussian concentration that there are at least $T - O(\log T)$ users have contexts and rewards bounded by $C_{1,T}, C_{2,T}$ with overwhelming probability, so we can still use above LDP mechanisms to protect their privacy. For users whose contexts and reward are out of range $C_{1,T}, C_{2,T}$, they can send a private version of $(\mathbf{0}, 0)$ vector so that there is no privacy leakage. We also impose regularize assumptions on the link function, which are common in previous work [44, 31, 39] and the corresponding family contains a lot of commonly-use model, e.g., linear model, logistic model.

**Assumption 1.** *The link function $\mu$ is continuously differentiable, Lipschitz and there exists some $\zeta > 0$ such that $\inf_{x \in [-C_B, C_B]} \mu'(x) = \zeta > 0$.*

## 3 Single-Parameter Setting

In this section, we develop a LDP contextual bandit framework (Algorithm 1) by combining statistical estimation and privacy mechanisms in the single-param bandit setting to achieve optimal regret bound in various cases. We use an abstract privacy mechanism $\psi$ in (1) and estimator $\varphi$ in (2) to allow the plug-in of various components.

### 3.1 Privacy Guarantee

For the linear case where the link function $\mu(x) = x$, we can use the following ordinary least square (OLS) estimator. Let with $\sigma_{\varepsilon,\delta} = 2\sqrt{2\ln(1.25/\delta)}/\varepsilon$. Define $M_t = x_{t,a_t} x_{t,a_t}^T + W_t$ where $W_t$ is a random matrix with $W_t(i,j) \sim \mathcal{N}(0, 4C_B^2 \sigma_{\varepsilon,\delta}^2)$ and $W_t(j,i) = W_t(i,j)$, and $u_t = r_t x_{t,a_t} + \xi_t$ where $\xi_t$ is a random vector following distribution $\mathcal{N}(0, C_B^2 c_r^2 \sigma_{\varepsilon,\delta}^2 I_d)$. The OLS privacy mechanism and the corresponding estimator are

$$\psi_t^{OLS}(x_{t,a_t}, r_t; \hat{\theta}_{t-1}) = (M_t, u_t), \tag{3}$$

$$\varphi_t^{OLS}(Z_1, \ldots, Z_t; \hat{\theta}_{t-1}) = \left( \sum_{i=1}^t M_i + \tilde{c}\sqrt{t}I \right)^{-1} \sum_{i=1}^t u_i, \tag{4}$$

where $\tilde{c} > 0$ is to be determined. We have the following LDP guarantee using the Gaussian mechanism (Lemma 2.1) and post-processing (Lemma 2.3).

**Algorithm 1:** LDP Single-parameter Contextual Bandit

---

**Input:** Time horizon $T$; Privacy Level $\varepsilon, \delta$.

1 **Initialization:** Setting $\hat{\theta}_0 = \mathbf{0}$.

2 **for** $t \leftarrow 1$ *to* $T$ **do**

3     **User side:**

4        Receive $\hat{\theta}_{t-1}$ from the server.

5        Pull arm $a_t = \text{argmax}_{a \in [K]} x_{t,a}^T \hat{\theta}_{t-1}$ and receive $r_t$.

6        Generate $Z_t$ by

$$Z_t = \psi_t(x_{t,a_t}, r_t; \hat{\theta}_{t-1}). \tag{1}$$

7     **Server side:**

8        Receive $Z_t$ from the user.

9        Update the estimation via

$$\hat{\theta}_t = \varphi_t(Z_1, \ldots, Z_t; \hat{\theta}_{t-1}). \tag{2}$$

10 **end**

---

**Proposition 3.1.** *Algorithm 1 with the private OLS update mechanism $\psi_t^{OLS}$ and estimator $\varphi_t^{OLS}$ is $(\varepsilon, \delta)$-LDP for $\varepsilon \in (0, 1]$.*

For the general link function $\mu$, its non-linearity adds to the difficulty in terms of both privacy-preserving and bandits. To estimate parameters in generalized linear bandits, one common approach to use a maximum likelihood estimator (MLE) at each step. In contrast to OLS solution, MLE does not have a close form solution with simple sufficient statistics in general. Thus, solving an MLE optimization procedure requires using all the previous data points and conducting costly operations at each round, resulting in time complexity and memory usage increasing with time. Instead, we use a one-step stochastic gradient approximation to incrementally update the estimator with the new observation at each round. To obtain a LDP version of this approximation, we use the LDP $l_2$-ball mechanism in Lemma 2.2.

$$\psi_t^{SGD}(x_{t,a_t}, r_t; \hat{\theta}_{t-1}) = \Psi_{\varepsilon, R}\left(\left(\mu(x_{t,a_t}^T \hat{\theta}_{t-1}) - r_t\right)x_{t,a_t}\right), \tag{5}$$

$$\varphi_t^{SGD}(Z_1, \ldots, Z_t; \hat{\theta}_{t-1}) = \hat{\theta}_{t-1} - \eta_t \psi_t^{SGD}. \tag{6}$$

where $\eta_t > 0$ is the stepsize to be determined and $R = 2c_r C_B$. Similarly, we can prove the following LDP guarrantee using the $l_2$-ball mechanism Lemma 2.2 and post-processing Lemma 2.3.

**Proposition 3.2.** *Algorithm 1 with the private SGD update mechanism $\psi_t^{SGD}$ and estimator $\varphi_t^{SGD}$ is $\varepsilon$-LDP.*

### 3.2 Regret Analysis

To derive the regret bound of our framework, we need the following assumptions on the marginal distribution $P_X$ of the stochastic contexts $\{x_{t,a}\}_{a \in [K]}$.

**Assumption 2.** *There exists some $\kappa_u > 0$ such that $\lambda_{\max}(\Sigma_a) \leq \frac{\kappa_u}{d}$ where $\Sigma_a$ is the covariance matrix of $P_X$ and $\lambda_{\max}(\Sigma_a)$ is the largest eigenvalues of $\Sigma_a$.*

**Assumption 3.** *For every $\|u\|_2 = 1$, denote $a^* = \arg\max_{a \in [K]} x_{t,a}^T u$, there exist some $\kappa_l > 0, p_* > 0$ such that $P_u((x^T v)^2 > \kappa_l/d) \geq p_*$ holds for any $u, v \in S_1^{d-1}$, where $P_u(\cdot)$ is the distribution of $x_{t,a^*}$.*

Similar assumptions are common in the analysis of single-parameter contextual bandits, e.g. [13, 22], and our conditions contain a wide range of distributions, including sub-gaussian with bounded density. See appendix A for discussion. Now we can show that our framework indeed achieves optimal regret bound.

**Theorem 3.1.** *Under Assumptions 2 and 3, with the choice of $\tilde{c} = 2\sigma_{\varepsilon, \delta}(4\sqrt{d} + 2\log(2T/\alpha))$ in (4), Algorithm 1 with OLS mechanism $\psi_t^{OLS}$ and estimator $\varphi_t^{OLS}$ achieve the following regret with*

*probability at least $1 - \alpha$ for some constant $C$,*

$$Reg(T) \leq C\sqrt{T}(C_B(\sigma_{\varepsilon,\delta} + \sigma_\epsilon)d\frac{\sqrt{(d + \log(T/\alpha))\log(KT/\alpha)}}{\kappa_l p_*} + o(1))$$

*Under Assumptions 1–3, with the choice of $\eta_t = c'd/(\kappa_l \zeta p_* t)$ for some $c' > 1$ in (6), Algorithm 1 with SGD mechanism $\psi_t^{SGD}$ and estimator $\varphi_t^{SGD}$ achieves the following regret with probability at least $1 - \alpha$ for some constant $C$,*

$$Reg(T) \leq C\sqrt{T}(\frac{r_{\varepsilon,d}\sqrt{d}}{\zeta\kappa_l p_*} \log\log(T/\alpha) + o(1)).$$

*with $o(1)$ means some factor that turns to 0 as $T \to \infty$.*

In the algorithm we shift the sample covariance matrix by $\tilde{c}\sqrt{t}$ to ensure the positive-definiteness of the noise matrix as in [34]. Such a shift guarantee the estimation accuracy in the early stage. Note that the optimal worst-case regret bound in the non-privacy case is $\tilde{O}(T^{1/2})$, our results show that we can achieve the same regret bound as in the non-privacy case in terms of time $T$. In fact, we can show a $\Omega(\sqrt{T}/\varepsilon)$ lower bound in this setting even when $K = 2$, which verified our optimal dependence on both $T$ and $\varepsilon$.

**Theorem 3.2.** *For $\theta \in \mathbb{R}^d$ and an algorithm $\pi$, we denote $\mathbb{E}[Reg_\pi(T; \theta)]$ the expectation regret of $\pi$ when the underlying parameter is $\theta$. When $K = 2$ and $x_{t,a} \sim \mathcal{N}(0, I_d/d)$ are independent over $a \in [K]$, we have for any possible $\varepsilon$-LDP algorithm $\pi$, $\sup_{\theta^*:\|\theta^*\|_2 \leq 1} \mathbb{E}[Reg_\pi(T; \theta^\star)] = \Omega(\sqrt{T}/\varepsilon)$.*

Given the best known $O(T^{3/4})$ regret bound of adversarial contextual LDP bandit in [44], our $O(\sqrt{T}/\varepsilon)$ result points out a possible gap between stochastic contextual bandits and adversarial contextual bandits under the LDP constraint.

The bounds given above are problem-independent, which do not dependent on the underlying parameters. If we consider an additional assumption that there is a gap between the optimal arm and the rest, which is usually the case when the number of contexts is small, then we can obtain sharper bounds than the problem-independent ones in Theorems 3.1.

**Assumption 4** (($\gamma, \beta$)-margin condition). *We say $P_X$ satisfies the $(\gamma, \beta)$-strong margin condition with $\gamma > 0, 0 < \beta \leq 1$, if for $\triangle_t := \mu(x_{t,a_t^*}^T \theta^\star) - \max_{j \neq a_t^*} \mu(x_{t,j}^T\theta^\star)$ and $h \in [0, b]$ with some positive constant $b$, we have $\mathbb{P}[\triangle_t \leq h] \leq \gamma h^\beta$.*

**Theorem 3.3.** *Under Assumptions 2–4 with the same choice of $\tilde{c}$ in Theorems 3.1, Algorithm 1 with OLS mechanism $\psi_t^{OLS}$ and estimator $\varphi_t^{OLS}$ achieves the following regret with probability at least $1 - \alpha$ for some constant $C$,*

$$Reg(T) \leq C \cdot \begin{cases} \gamma C_B \log T[(\frac{C_B d(C_B\sigma_\epsilon + \sigma_{\varepsilon,\delta})\sqrt{d + \log(T/\alpha)}}{\kappa_l p_*})^2 + o_{\beta,\gamma}(1)], & \beta = 1, \\ \frac{\gamma C_B}{1 - \beta}T^{\frac{1-\beta}{2}}[(\frac{C_B d(C_B\sigma_\epsilon + \sigma_{\varepsilon,\delta})\sqrt{d + \log(T/\alpha)}}{\kappa_l p_*})^{1+\beta} + o_{\beta,\gamma}(1)], & 0 \leq \beta < 1. \end{cases}$$

*Under Assumptions 1–4 and with the same choice of $\eta_t$ in Theorems 3.1, Algorithm 1 with SGD mechanism $\psi_t^{SGD}$ and estimator $\varphi_t^{SGD}$ achieves the following regret with probability at least $1 - \alpha$ for some constant $C$,*

$$Reg(T) \leq C \cdot \begin{cases} \gamma LC_B \log T[(\frac{r_{\varepsilon,d}LdC_B\sqrt{\log(\log(T)/\alpha)}}{\zeta\kappa_l p_*})^2 + o_{\beta,\gamma}(1)], & \beta = 1, \\ \frac{\gamma LC_B}{1 - \beta}T^{\frac{1-\beta}{2}}[(\frac{r_{\varepsilon,d}LdC_B\sqrt{\log(\log(T)/\alpha)}}{\zeta\kappa_l p_*})^{1+\beta} + o_{\beta,\gamma}(1)], & 0 \leq \beta < 1. \end{cases}$$

*with $o_{\beta,\gamma}(1)$ being a factor depending on $\beta, \gamma$ that converges to 0 as $T \to \infty$.*

Unlike in the worst-case bound, it is more challenge to establish corresponding lower under the margin condition. A possible roadmap to show the $\Omega(\log T/\varepsilon^2)$ lower bound is to follow the argument in [21]: Their argument uses the Van-Tree inequality to bound the mean squared estimation error from below for each time to show the $\Omega(\log T)$ bound in non-private setting. To consider the influence of private noise, it is possible to combine the LDP-version Van-Tree inequality in [2] with the above argument to get $\Omega(\log T/\varepsilon^2)$ bound.

## 4 Multi-parameter Setting

In this section, we present our LDP framework for the multiple parameter setting. Compared with the single parameter setting, this framework introduces three non-trivial components to match classical regret bounds while still guarantee LDP: warm up, synthetic update and elimination.

---

**Algorithm 2:** LDP Multi-parameter Contextual Bandit

---

**Input:** Time horizon $T$; Warm up period length $s_0$; Privacy Level $\varepsilon, \delta$.

1 **Initialization:** Setting $\hat{\theta}_{0,i} = 0, i \in [K]$.
2 **for** $t \leftarrow 1$ *to* $Ks_0$ **do**
3     **User side:**
4        Receiving $\hat{\theta}_{t-1,1:K}$ from the server.
5        Pulling arm $a_t := (t \bmod K) + 1$ and receive $r_t$.
6        Generate and update $Z_{t,i} = \mathbf{1}\{a_t = i\}\psi_t(X_t, r_t; \hat{\theta}_{t-1,i}), i \in [K]$ to the server.
7     **Server side:**
8        Receive the update $Z_{t,1:K}$ from the user.
9        Re-estimate parameters via $\hat{\theta}_{t,i} := \varphi_t(Z_{1,i}, \ldots, Z_{t,i}), \forall i \in [K]$.
10 **end**
11 **for** $t \leftarrow Ks_0 + 1$ *to* $T$ **do**
12     **User side:**
13        Receive $\hat{\theta}_{t-1,1:K}$ from the server.
14        Determine a subset $\hat{K}_t$ of $[K]$ by setting

$$\hat{K}_t := \{a \in [K] : X_t^T \hat{\theta}_{Ks_0,a} > \max_{a \in [K]} X_t^T \hat{\theta}_{Ks_0,a} - \frac{h}{2}\} \tag{7}$$

15        Pulling arm $a_t := \operatorname{argmax}_{a \in \hat{K}_t} \mu(X_t^T \hat{\theta}_{t-1,a})$ and receive $r_t$.
16        Generating information for all arms $\{Z_{i,t}\}_{i \in [K]}$ by setting

$$Z_{i,t} = \begin{cases} \psi_t(X_t, r_t; \hat{\theta}_{t-1,i}) & \text{if } a_t = i, \\ \psi_t(\mathbf{0}, 0; \hat{\theta}_{t-1,i}) & \text{otherwise.} \end{cases}$$

17     **Server side:**
18        Receive the update $\{Z_{i,t}\}_{i \in [K]}$ from the user.
19        Re-estimate parameters via

$$\hat{\theta}_{t,i} := \varphi_t(Z_{1,i}, \ldots, Z_{t,i}).$$

20 **end**

---

**Warm up.** In the warm up stage, all arms are given equal opportunities to be explored for a preliminary estimation of their parameters. Such estimation does not aim for the accuracy to select the optimal arm with high probability. Instead, we only need accuracy at the level of ruling out the substantially inferior arms. Thus, this stage only needs $O(\log T)$ steps.

Since the actions in this stage are independent of the contexts, there is no need to protect the pulled arm. However, we still need to protect the contexts by using a privacy mechanism similar in the single-parameter setting.

**Synthetic update.** After the warm up, we need to make decisions based on the contexts to achieve vanishing regret. In order to obtain the privacy guarantee, we introduce our synthetic update mechanism. Although in each time only one arm is pulled, we create synthetic data for all unselected arms. In this way, the server receives synthetic feedback about all arms, regardless of whether it is selected or not, and thus cannot figure out which one is selected.

Another method to provide LDP protection for the selected arm is to ensure the action $a_t$ satisfies LDP. However, the regret will grow linearly, as shown in [34].

**Elimination.** We use the information obtained during warm up to exclude obviously inferior arms. Such a method has been applied in [5] to guarantee a certain kind of independence of the information in each round. However, we use this method for a different purpose. The necessity of such an elimination strategy comes from protecting privacy in the multi-parameter setting. Although we have obtained an estimation to a certain level of accuracy in the warm up stage, our knowledge on un-selected arms will be gradually corrupted by the noise incurred in the synthetic update in each round. Such corruption will make us fail to distinguish arms that are possibly optimal from the surely sub-optimal ones. To avoid corruption, we may need to pick the sub-optimal arms frequently but this will result in large regret. That is why we use the warm up information to eliminate the arms with extremely poor performance as in (7).

## 4.1 Privacy Guarantee

The OLS/SGD mechanisms and estimators are the same as (3)–(6) in the single-parameter setting. To prevent the server from distinguishing the selected arm from the other $K - 1$ arms, a straightforward idea is to use $(\varepsilon/K, \delta/K)$-LDP mechanism for the synthetic update by composition property in lemma 2.4. However, we can prove that our algorithm can still achieve the same LDP guarantee with a much less stringent privacy mechanism, say $(\varepsilon/2, \delta/2)$-LDP, in Propositions 4.1 and 4.2.

**Proposition 4.1.** *Algorithm 2 with the private OLS update mechanism $\psi_t^{OLS}$ and estimator $\varphi_t^{OLS}$ is $(\varepsilon, \delta)$-LDP.*

**Proposition 4.2.** *Algorithm 2 with the private SGD update mechanism $\psi_t^{SGD}$ and estimator $\varphi_t^{SGD}$ is $\varepsilon$-LDP.*

## 4.2 Regret Analysis

**Assumption 5** (Diversity condition)**.** *Let $K_{opt}$ and $K_{sub}$ be a partition of $[K]$ such that for any $i \in K_{sub}$, $\mu(X^T\theta_i) < \max_{j \neq i} \mu(X^T\theta_j) - h_{sub}$ for some $h_{sub} > 0$ and every $X \in \mathcal{X}$. For any $i \in K_{opt}$ define the set $U_i := \{X : \mu(X^T\theta_i) > \max_{j \neq i} \mu(X^T\theta_j)\}$. There exists $\kappa_l > 0, p' > 0$ such that for all $i \in K_{opt}$ and unit vector $v, \mathbb{P}((v^T X)^2 \mathbf{1}\{X \in U_i\} \geq \kappa_l/K_{opt}) > p'$.*

**Assumption 6** (($\gamma, \beta$)-margin condition)**.** *This is almost identical to Assumption 4 except that we replace $\triangle_t$ with $\triangle_t := \mu(X_t^T \theta_{a_t^*}) - \max_{j \neq a_t^*} \mu(X_t^T \theta_j)$.*

In our algorithm, diversity condition guarantees that conditioning on the arm $i$ is pulled, the distribution of $X_t$ still can provide enough information about $\theta_i$. We would remark here that we need no longer any deterministic gap in the definition of $U_i$, which weakens the assumption made in [4],[5]. Now we are in the suited position to present our theoretical guarantee of the algorithm.

**Theorem 4.1.** *Under Assumptions 1, 5 and 6, with the choice of $\tilde{c} = 2\sigma_{\varepsilon/2, \delta/2}(4\sqrt{d} + 2\log(2TK/\alpha))$ in (4), $s_0 = C \cdot K(\frac{C_B \sigma_\epsilon + \sigma_{\varepsilon, \delta}}{\min\{\lambda_0, h\} p' \kappa_l})^2(d + \log(TK/\alpha))$ and $h = h_{sub}, \lambda_0 = (2\gamma L C_B)^{-1}(\frac{p'}{2})^{1/\beta}$, Algorithm 2 with OLS mechanism $\psi_t^{OLS}$ and estimator $\varphi_t^{OLS}$ achieve the following regret with probability at least $1 - \alpha$ for some constant $C$,*

$$Reg(T) \leq \gamma C C_B \Big[ \Big( \frac{K C_B (C_B \sigma_\epsilon + \sigma_{\varepsilon, \delta}) \sqrt{d + \log((TK)/\alpha)}}{\kappa_l p'} \Big)^{1+\beta} + o_{h_{sub}, \beta, \gamma}(1) \Big] \cdot \begin{cases} \log T, & \beta = 1, \\ \frac{T^{\frac{1-\beta}{2}}}{1 - \beta}, & 0 < \beta < 1. \end{cases}$$

*Under Assumptions 1, 5 and 6, with the choice of step-size*

$$\eta_t := (\mathbf{1}_{\{t \leq K s_0\}}((t \bmod K) + 1) + \mathbf{1}_{\{t > K s_0\}}(t - (K-1)s_0))^{-1} K_{opt}(\zeta \kappa_l p')^{-1} c'$$

*for any $c' \geq 1$ and $h = h_{sub}$, Algorithm 2 with SGD mechanism $\psi_t^{SGD}$ and estimator $\varphi_t^{SGD}$ achieve the following regret with probability at least $1 - \alpha$ for some constant $C$,*

$$Reg(T) \leq \gamma L C C_B \Big[ \Big( \frac{K r_{\varepsilon, d} L C_B \sqrt{\log((TK \log T)/\alpha)}}{\zeta \kappa_l p'} \Big)^{1+\beta} + o_{h_{sub}, \beta, \gamma}(1) \Big] \cdot \begin{cases} \log T, & \beta = 1, \\ \frac{T^{\frac{1-\beta}{2}}}{1 - \beta}, & 0 < \beta < 1. \end{cases}$$

Theorem 4.1 recovers the non-privacy bound in [5] under similar condition up to a logarithmic factor. Notice that unlike Theorem 3.3 in the single-parameter case, we cannot establish the regret when

$\beta = 0$. The reason is that in our analysis, we need the probability of $\triangle_t > h$ vanish as $h \to 0$ to guarantee the estimation error for $\theta_i, i \in K_{opt}$ converges. The corresponding theoretical result in this setting when $\beta = 0$ is left as an open question.

## 5 Experiment

To the best of our knowledge, the contextual bandit algorithms with LDP guarantee has only been studied by [44], who propose a variant of LinUCB algorithm for linear bandits and a variant of Generalized Linear Online-to-confidence-set Conversion (GLOC) framework [23] for generalized linear bandits. We refer their methods as LDP-UCB and LDP-GLOC. We call our method LDP-OLS if we plug in the OLS mechanism and estimator into Algorithms 1 and 2, and LDP-SGD if we plug in the SGD ones. We evaluate all the four methods on two different privacy levels $\varepsilon = 0.5$ and 1 in synthetic datasets, which are industry standards. For example, Apple uses $\varepsilon = 4$ in their projects on Emojis and Safari usage [38]. Similar choices of the privacy parameter $\varepsilon$ can be found in [3, 20]. We also demonstrate the efficacy of our algorithms with real data on Auto Lending[2] in Appendix G. For the sake of comparison, the learning step parameter for LDP-GLOC and LDP-SGD are tuned in the same way.[3] The first and second columns in Figure 4 are for single-param and multi-param settings, respectively, which are simulation studies on linear bandits. The context is generated from $\text{Unif}(S_1^{d-1})$ at each round.

In conclusion, our methods significantly outperform existing ones in all settings consistently. LDP-SGD achieves better performance under more strigent privacy requirements.

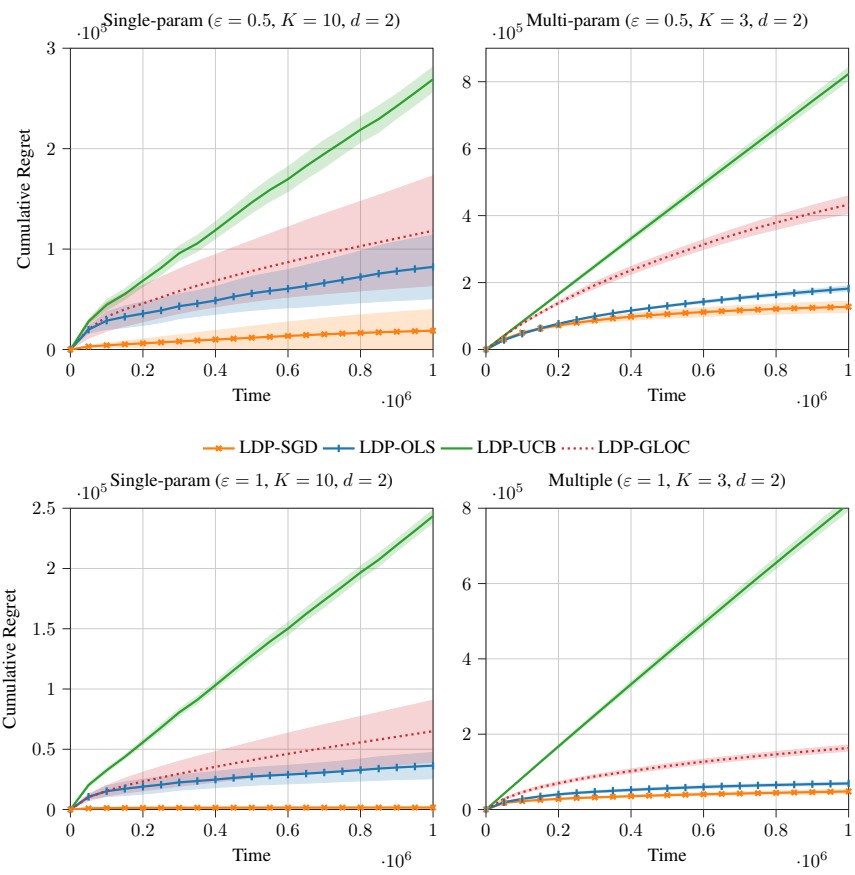

Figure 2: We perform 10 replications for each case and plot the mean and 0.5 standard deviation of their regrets.

---

[2]On-Line Auto Lending dataset CRPM-12-001 provided by Columbia University `https://www8.gsb.columbia.edu/cprm/research/datasets`, and has been used in the study of contextual bandits by [26, 11].

[3]The source code to reproduce all the results is available at the GitHub repo liangzp/LDP-Bandit.

# 6    Conclusion

In this paper, we propose LDP contextual bandit frameworks in both single-parameter and multi-parameter settings with flexibility to deal generalized linear reward structure, and establish theoretical guarantee of our algorithms based on the frameworks. Our algorithms are highly efficient and have superior empirical performance. There are still some open questions to be explored. Whether our regret bounds are optimal in terms of $\varepsilon$ in the multi-parameter setting is still unknown. It will be interesting to explore estimators and mechanisms beyond the private OLS and SGD ones to study the optimality in terms of $\varepsilon$. Moreover, whether there is a fundamental limit in adversarial contextual bandit under LDP constraints is still an open question. It also remains an open question to analyze the regret bound in the multi-parameter setting when $\beta = 0$ in the margin condition.

## Acknowledgments and Disclosure of Funding

This work was supported by Hong Kong Research Grant Council (Grants 16317416, 16204718, 16214121) and Guangdong-Hong Kong-Macao Joint Laboratory for Data-Driven Fluid Mechanics and Engineering Applications.

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
