# A  Randomness Condition

In this section, we show that a sub-gaussian random vector with bounded density satisfies Assumption 3:

We say a random vector $x$ is $\sigma^2$-sub-gaussian vector with bounded density, if for every $v \in S_1^{d-1}$, $v^T x$ is $\sigma^2$-sub-gaussian and its density function exists and is bounded by $\gamma$ for some $\gamma > 0$. For such kind of random vector, [30] shows that it satisfies Assumption 3 with $\kappa_l = \dfrac{2d}{3\gamma K}$ and $p_* = \dfrac{1}{3}$. In particular, [22] shows that when $x$ follows $\mathcal{N}(0, \Sigma)$, with $\lambda_{\min}(\Sigma) \geq \dfrac{\kappa}{d}$, we can have $\kappa_l = \dfrac{c_1 \kappa}{d}$ and $p_* = c_2$ for constants $c_1$ and $c_2$.

# B  Proof of Privacy Guarantee

## B.1  Proof of Results in Section 3.1

*Proof of Proposition 3.1.* Since we assume that the features and rewards are bounded, $\|x_{t,a}\| \leq C_B, \|r_t\| \leq c_r$ for all $t \in [T]$ and $a \in [K]$, by Lemma 2.1, $M_t$ is $(\varepsilon/2, \delta/2)$-LDP and $u_t$ is $(\varepsilon/2.\delta/2)$-LDP. Thus Lemma 2.4 implies that $\psi_t^{OLS}$ is $(\varepsilon, \delta)$-LDP. $\qquad\square$

*Proof of Proposition 3.2.* Since we assume that the features and rewards are bounded, $\|x_{t,a}\| \leq C_B, \|r_t\| \leq c_r$ for all $t \in [T]$ and $a \in [K]$, we have $(\mu(x_{t,a_t}^T \hat{\theta}_{t-1}) - r_t) x_{t,a_t}$ bounded by $2c_r C_B$. Lemma 2.2 implies that $\psi_t^{SGD}$ is $\varepsilon$-LDP. $\qquad\square$

## B.2  Proof of Results in Section 4.1

*Proof of Proposition 4.1.* We simply denote $\psi_t^{OLS}$ by $\psi_t$ in this proof. At time $t$, for any two $x \neq x'$, without loss of generality assuming the action corresponding $x$ and $x'$ are $a_t = 1$ and $a_t = 2$, then the output corresponding $x, x'$ is given by $(\psi_t(x, x^T\theta_1 + \epsilon_t), \psi_t(0,0), \ldots, \psi_t(0,0))$ and $(\psi_t(0,0), \psi_t(x', x'^T\theta_2 + \epsilon_t), \ldots, \psi_t(0,0))$. Since $\psi_t(0,0)$ has the same distribution, we have for any subset $A_1 \times A_2 \times \cdots \times A_K \subset \mathbb{R}^{Kd}$ with $A_i$ a Borel set in $\mathbb{R}^d$,

$$
\begin{aligned}
&\frac{\mathbb{P}(\psi_t(x, x^T\theta_1 + \epsilon_t) \in A_1, \psi_t(0,0) \in A_2, \ldots, \psi_t(0,0) \in A_K)}{\mathbb{P}(\psi_t(0,0) \in A_1, \psi_t(x', x'^T\theta_2 + \epsilon_t) \in A_2, \ldots, \psi_t(0,0) \in A_K)} \\
&= \frac{\mathbb{P}(\psi_t(x, x^T\theta_1 + \epsilon_t) \in A_1, \psi_t(0,0) \in A_2)}{\mathbb{P}(\psi_t(x', x'^T\theta_2 + \epsilon_t) \in A_2, \psi_t(0,0) \in A_1)}.
\end{aligned} \tag{8}
$$

Set $\tilde{\psi}(v_1, v_2) := (\psi_t(v_1), \psi_t(v_2))$, and $(v_1, v_2) := (x, 0), (v_1', v_2') := (0, x')$, then we have (8) equals to $\tilde{\psi}(v_1, v_2)/\tilde{\psi}(v_1', v_2')$, thus applying Lemma 2.4 to it implies that (8) is upper bounded by $e^\varepsilon + \delta \mathbb{P}(\psi_t(x', x'^T\theta_2 + \epsilon_t) \in A_2, \psi_t(0,0) \in A_1)^{-1}$, leading to the desired result. $\qquad\square$

*Proof of Proposition 4.2.* That is nearly the same as the proof of Proposition 4.1, but replacing $e^\varepsilon + \delta \mathbb{P}(\psi_t(x', x'^T\theta_2 + \epsilon_t) \in A_2, \psi_t(0,0) \in A_1)^{-1}$ by $e^\varepsilon$ in the last step. $\qquad\square$

# C  Proof of Results in Section 3.2

In the following analysis, without special explaination, all the $c$ and $C$ denote absolute constants. Sometimes we state the inequality of type $A_1 \leq C \log(A_2/\alpha) A_3$ holds with probability at least $1 - \alpha$ while in proof we derive the results hold with $1 - c\alpha$ for some constant c. In fact, they are equivalent by re-scaling $\alpha$ and changing $C$ to some larger constant.

## C.1  Proof of Worst-Case Bounds

*Proof of Theorem 3.1.* Since $x_{t,a_t}$ is the greedy selection, we have $x_{t,a_t}^T \hat{\theta}_{t-1} \geq x_{t,a}^T \hat{\theta}_{t-1}$ for any time $t \in [T]$ and $a \in [K]$. Consequently we have the following upper bound for the instantaneous

regret at time $t$,

$$\max_{a\in[K]} (x_{t,a} - x_{t,a_t})^T \theta^\star \leq \max_{a\in[K]} (x_{t,a} - x_{t,a_t})^T \left(\theta^\star - \hat{\theta}_{t-1}\right)$$

$$\leq \max_{a,a'\in[K]} (x_{t,a} - x_{t,a'})^T \left(\theta^\star - \hat{\theta}_{t-1}\right)$$

$$\leq 2 \max_{a\in[K]} \left| x_{t,a}^T \left(\theta^\star - \hat{\theta}_{t-1}\right) \right|.$$

For any fixed $a \in [K]$, $x_{t,a}$ is independent of $\hat{\theta}_{t-1}$. By Assumption 3, conditioning on the historical information up to time t, $x_{t,a}^T(\theta^\star - \hat{\theta}_{t-1})$ is a $\frac{\kappa_u}{d}\|\theta^\star - \hat{\theta}_{t-1}\|^2$-sub-gaussian random variable. Now by the maximal concentration inequality for a sub-gaussian sequence, we have with probability at least $1 - \frac{\alpha}{T}$,

$$\max_{a\in[K]} |x_{t,a}^T(\theta^\star - \hat{\theta}_{t-1})| = O\left(\sqrt{\frac{\kappa_u \log(KT/\alpha)}{d}} \|\theta^\star - \hat{\theta}_{t-1}\|\right).$$

To control the regret bound, we bound the estimation error $\|\theta^\star - \hat{\theta}_{t-1}\|$ in each time in the following lemma.

**Lemma C.1** (Estimation Error for OLS). *Using the private OLS update mechanism $\psi_t^{OLS}$ and estimator $\varphi_t^{OLS}$, for any $8\frac{d \log 9 + \log(T/\alpha)}{p_*^2} < t \leq T$, we have with probability at least $1 - \frac{\alpha}{T}$,*

$$\|\hat{\theta}_t - \theta^\star\|^2 \leq C(C_B \sigma_\epsilon \sigma_{\varepsilon,\delta} d)^2 \frac{d + \log(T/\alpha)}{\kappa_l^2 p_*^2 t}, \tag{9}$$

*for some C independent of d, K and T.*

**Lemma C.2** (Estimation Error for SGD). *Using the private OLS update mechanism $\psi_t^{SGD}$ and estimator $\varphi_t^{SGD}$, for any $3 \leq t \leq T$, we have with probability at least $1 - \frac{\alpha}{T}$,*

$$\|\hat{\theta}_t - \theta^\star\|^2 \leq \frac{(624 \log(\log T/\alpha) + 1)r_{\varepsilon,d}^2 d^2}{4\kappa_l^2 \zeta^2 p_*^2 t}. \tag{10}$$

Plugging OLS estimation error (9) into the regret bound, denote $t_1 := 8\frac{d \log 9 + \log(T/\alpha)}{p_*^2}$, the following holds with probability at least $1 - \alpha$,

$$\sum_{t=1}^T \max_{a\in[K]} (x_{t,a} - x_{t,a_t})^T \theta^\star$$

$$\leq t_1 c_r + \sum_{t=t_1+1}^T CC_B \sigma_\epsilon \sigma_{\varepsilon,\delta} d \sqrt{\frac{\kappa_u \log(KT/\alpha)}{d}} \frac{\sqrt{d + \log(T/\alpha)}}{\kappa_l p_* \sqrt{t}} \tag{11}$$

$$\leq 8\frac{d \log 9 + \log(T/\alpha)}{p_*^2} + CC_B \sigma_{\varepsilon,\delta} \sigma_\epsilon \sqrt{d} \frac{\sqrt{d + \log(T/\alpha)}}{\kappa_l p_*} \sqrt{\kappa_u \log(KT/\alpha)} \sqrt{T}.$$

Plugging the SGD estimation error (10) into the regret bound, we have

$$\sum_{t=1}^T \max_{a\in[K]} (x_{t,a} - x_{t,a_t})^T \theta^\star$$

$$\leq 2c_r + \sum_{t=3}^T \sqrt{\kappa_u \log(KT/\alpha)} \frac{\sqrt{(624 \log(\log T/\alpha) + 1)} r_{\varepsilon,d} \sqrt{d}}{2\kappa_l \zeta p_* \sqrt{t}}$$

$$\leq 2c_r + \frac{\sqrt{(624 \log(\log T/\alpha) + 1)} r_{\varepsilon,d} \sqrt{d}}{2\zeta \kappa_l p_*} \sqrt{\kappa_u \log(KT/\alpha)} \sqrt{T}. \tag{12}$$

$\square$

So now it suffices to prove the Lemmas C.1 and C.2.

## C.2 Proof of lemma C.1

**Lemma C.3.** *As long as $t > 8\frac{d\log 9 + \log(T/\alpha)}{p_*^2}$, the following lower bound*

$$\lambda_{\min}(\sum_{i=1}^{t} x_{i,a_i} x_{i,a_i}^T) \geq C \cdot \frac{t\kappa_l p_*}{d},$$

*holds with probability at least $1 - \frac{\alpha}{T}$, for some $C$ independent of $d$ and $T$.*

*Proof.* Define $\mathcal{F}_t^-$ as the filtration generated by $\{x_{i,a_i}\}_{i \in [t-1]}$, $\{\epsilon_i\}_{i \in [t-1]}$ and the randomness from $\{\psi_i^{OLS}\}_{i \in [t-1]}$. By greedy algorithm, in each time $i$, $x_{i,a_i}$ is selected as $a_i = \text{argmax}_{a \in [K]} x_{i,a}^T \hat{\theta}_{i-1}$. Thus by the Assumption 3, we have for any $0 < s < p_*$,

$$\mathbb{P}(\sum_{i=1}^{t}(x_{i,a_i}^T v)^2 < t\kappa_l(p_* - s)/d)$$

$$\leq \mathbb{P}(\sum_{i=1}^{t} \mathbf{1}\{(x_{i,a_i}^T v)^2 > \kappa_l/d\} < t(p_* - s))$$

$$\leq \mathbb{P}(\frac{1}{t}\sum_{i=1}^{t}(\mathbf{1}\{(x_{i,a_i}^T v)^2 > \kappa_l/d\} - \mathbb{E}[\mathbf{1}\{(x_{i,a_i}^T v)^2 > \kappa_l/d\}|\mathcal{F}_i^-])) < -s)$$

$$\leq \exp(-\frac{s^2 t}{2}),$$

where in the last inequality we use the Azuma–Hoeffding's inequality for bounded martingale-difference sequence (see Corollary 2.20 in [42]).

For every $d \times d$ positive-definite matrix $A$, with an abuse of notation, we denote $\mathcal{N}_\varepsilon$ as the $\varepsilon$-net of $S_1^{d-1}$ for some $\varepsilon > 0$ to be determined,

$$\lambda_{\max}(A) \leq \frac{1}{1 - 2\varepsilon} \sup_{x \in \mathcal{N}_\varepsilon} x^T A x,$$

which then implies

$$\lambda_{\min}(A) = -\lambda_{\max}(-A) \geq \frac{-1}{1 - 2\varepsilon} \sup_{x \in \mathcal{N}_\varepsilon} x^T(-A)x = \frac{1}{1 - 2\varepsilon} \inf_{x \in \mathcal{N}_\varepsilon} x^T A x.$$

By choosing $\varepsilon = 1/4$, we can find an $\varepsilon$-net $\mathcal{N}_\varepsilon$ with cardinality $|\mathcal{N}_\varepsilon| \leq 9^d$. Therefore

$$\lambda_{\min}(A) \geq 2 \inf_{x \in \mathcal{N}_\varepsilon} x^T A x.$$

Note that

$$\mathbb{P}(\min_{\|v\|=1} \sum_{i=1}^{t}(x_{i,a_i}^T v)^2 < 2t\kappa_l(p_* - s)/d) \leq \mathbb{P}(\sum_{i=1}^{t}(x_{i,a_i}^T v)^2 < t\kappa_l(p_* - s)/d, \exists v \in \mathcal{N}_\varepsilon)$$

$$\leq 9^d \exp(-\frac{s^2 t}{2}).$$

By setting $s = \sqrt{\frac{2d\log 9 + 2\log(T/\alpha)}{t}}$, we have when $t > 8\frac{d\log 9 + \log(T/\alpha)}{p_*^2}$ with probability at least $1 - \frac{\alpha}{T}$,

$$\lambda_{\min}(\sum_{i=1}^{t} x_{i,a_i} x_{i,a_i}^T) = \min_{\|v\|=1} \sum_{i=1}^{t} \langle x_{i,a_i}, v\rangle^2 \geq \frac{\kappa_l p_* t}{d}.$$

$\square$

*Proof of Lemma C.1.*

By lemma C.3 we know that with probability at least $1 - \frac{\alpha}{T}$,

$$\lambda_{\min}(\sum_{i=1}^{t} x_{i,a_i} x_{i,a_i}^T) \geq C_1 \kappa_l p_* t/d,$$

for some $C_1$ independent of $d, K$ and $T$.

Since $\{W_i\}_{i \in [t]}$ are independent, therefore by concentration bounds for Wigner matrix we have with probability at least $1 - \frac{\alpha}{T}$,

$$\|\sum_{i=1}^{t} W_i\|^2 \leq C_2 t \sigma_{\varepsilon,\delta}^2 (d + \log(T/\alpha)),$$

for some $C_2$ independent of $d, K$ and $T$. However, it is important to note that the perturbation of privacy noise matrix $\sum_{i=1}^{t} W_i$ may destroy the positive definite property of the Gram matrix $\sum_{i=1}^{t} x_{i,a_i} x_{i,a_i}^T$ when t is still small. Therefore, we shift $\sum_{i=1}^{t} W_i$ by adding $\tilde{c}\sqrt{t}I_d$ where $\tilde{c} := C_2 \sigma_{\varepsilon,\delta}(\sqrt{d} + \sqrt{\log(T/\alpha)})$.

We denote $A_t := \sum_{i=1}^{t}(x_{i,a_i} x_{i,a_i}^T + W_i) + \tilde{c}\sqrt{t}I$. Therefore, by Weyl's inequality we have with probability at least $1 - \frac{\alpha}{T}$,

$$\lambda_{\min}(A_t) = \lambda_{\min}\left(\sum_{i=1}^{t}(x_{i,at_i} x_{i,a_i}^T + W_i) + \tilde{c}\sqrt{t}I_d\right) \geq \lambda_{\min}\left(\sum_{i=1}^{t} x_{i,a_i} x_{i,a_i}^T\right) \geq C_1 \kappa_l p_* t/d.$$

So now we we study the OLS estimator with $x_{i,a_i}, \epsilon_i$ given above and $r_i = x_{i,a_i}^T \theta^\star + \epsilon_i$. In that case, the estimation error of the OLS estimator under LDP constraints at time $t$ is given by

$$\hat{\theta}_t - \theta^\star = A_t^{-1} \sum_{i=1}^{t}(x_{i,a_i} r_i + \xi_i) - \theta^\star$$

$$= A_t^{-1} \sum_{i=1}^{t}(x_{i,a_i} x_{i,a_i}^T \theta^\star + x_{i,a_i} \epsilon_i + \xi_i) - \theta^\star$$

$$= A_t^{-1}(\sum_{i=1}^{t} x_{i,a_i} \epsilon_i) - A_t^{-1} \sum_{i=1}^{t} W_i \theta^\star + A_t^{-1} \sum_{i=1}^{t} \xi_i - \tilde{c}\sqrt{t}A_t^{-1}\theta^\star.$$

Define $\mathcal{F}_t$ as the filtration generated by $\{x_{i,a_i}\}_{i \in [t]}, \{\epsilon_i\}_{i \in [t-1]}$ and the randomness from $\{\psi_i\}_{i \in [t-1]}$. Notice that for every unit vector $u$,

$$\mathbb{E}[\exp(\lambda \sum_{i=1}^{t} u^T x_{i,a_i} \epsilon_i)] = \mathbb{E}[\mathbb{E}[\exp(\lambda \sum_{i=1}^{t} u^T x_{i,a_i} \epsilon_i)|\mathcal{F}_t]]$$

$$= \mathbb{E}[\prod_{i=1}^{t-1} \exp(\lambda u^T x_{i,a_i} \epsilon_i)\mathbb{E}[\exp(\lambda u^T X_i \epsilon_i)|\mathcal{F}_t]]$$

$$\overset{(1)}{\leq} \exp(\frac{\lambda^2 C_B^2 \sigma_\epsilon^2}{2})\mathbb{E}[\prod_{i=1}^{t-1} \exp(\lambda u^T x_{i,a_i} \epsilon_i)]$$

$$\overset{(2)}{\leq} \exp(\frac{\lambda^2 C_B^2 \sigma_\epsilon^2 t}{2}).$$

Inequality (2) is due to the mathematical induction using the same technique in the equality (1). Thus $\sum_{i=1}^{t} x_{i,a_i} \epsilon_i$ is $\sigma^2 C_B^2 t$-sub-gaussian vector, and by the concentration of norm for sub-gaussian vectors, we have then with probability at least $1 - \frac{\alpha}{T}$,

$$\|\sum_{i=1}^{t} x_{i,a_i} \epsilon_i\|^2 \leq C_3 \sigma_\epsilon^2 C_B^2 t(d + \log(T/\alpha)),$$

where $C_3$ is a positive constant independent of $d$, $K$ and $T$.

Therefore,

$$
\begin{aligned}
\|A_t^{-1}(\sum_{i=1}^{t} x_{i,a_i}\epsilon_i)\|^2 &\leq \|A_t^{-1}\|^2\|(\sum_{i=1}^{t} x_{i,a_i}\epsilon_i)\|^2 \\
&\leq \frac{C_3\sigma^2 C_B^2 d^2 t(d+\log(T/\alpha))}{(C_1\kappa_l p_* t)^2}.
\end{aligned}
\tag{13}
$$

Moreover,

$$
\begin{aligned}
\|A_t^{-1}\sum_{i=1}^{t} W_i\theta^\star\|^2 &\leq \|A_t^{-1}\|^2\|\sum_{i=1}^{t} W_i\|^2\|\theta^\star\|^2 \\
&\leq \|A_t^{-1}\|^2\|\sum_{i=1}^{t} W_i\|^2 \\
&\leq \frac{C_2 t\sigma_{\varepsilon,\delta}^2(d+\log(T/\alpha))}{(C_1\kappa_l p_* t)^2},
\end{aligned}
\tag{14}
$$

where the second inequality is from the assumption that $\|\theta^\star\| \leq 1$.

Third, Since $\xi_i$ are random vector with independent, sub-gaussian coordinates that satisfy $\mathbb{E}\xi_{i,j}^2 = \sigma_{\varepsilon,\delta}^2$, $\sum_{i=1}^{t} \xi_i$ is a random vactor with independent sub-gaussian coordinates that satisfy $\mathbb{E}\sum_{i=1}^{t} \xi_{i,j}^2 = t\sigma_{\varepsilon,\delta}^2$. Therefore for all $t \in [T]$, with probability at least $1 - \frac{\alpha}{T}$,

$$
\|\sum_{i=1}^{t} \xi_i\|^2 \leq C_4 t\sigma_{\varepsilon,\delta}^2(d+\log(T/\alpha)),
$$

for some positive constant $C_4$ independent of $d$, $K$ and $T$. Therefore,

$$
\|A_t^{-1}\sum_{i=1}^{t} \xi_i\|^2 \leq \frac{C_4 t\sigma_{\varepsilon,\delta}^2 d^2(d+\log(T/\alpha))}{(C_2\kappa_l p_* t)^2}.
\tag{15}
$$

Lastly,

$$
\|\tilde{c}\sqrt{t}A_t^{-1}\theta^\star\|^2 \leq \frac{\tilde{c}^2 t}{(C_2\kappa_l p_* t)^2},
\tag{16}
$$

holds with probability at least $1 - \frac{\alpha}{T}$. Plugging all bounds (13) (14) (15) and (16) together we get then with probability at least $1 - \frac{\alpha}{T}$,

$$
\|\hat{\theta}_t - \theta^\star\|^2 \leq C_5\sigma_\epsilon^2 C_B^2\sigma_{\varepsilon,\delta}^2 d^2\frac{d+\log(T/\alpha)}{\kappa_l^2 p_*^2 t},
$$

for some positive constant $C_5$ independent of $d$, $K$ and $T$. $\qquad\square$

*Proof of Lemma C.2.* Denote $g_t$ as the gradient at time t, $\hat{g}_t := \Psi_\varepsilon[(\mu(x_{t,a_t}^T\hat{\theta}_t) - r_t)x_{t,a_t}]$ is the LDP private estimator of $g_t$ and $\hat{z}_t = g_t - \hat{g}_t$. By the unbiasedness of $\Psi_\varepsilon$ in Lemma 2.2 we have

$$
\begin{aligned}
&\mathbb{E}[\Psi_\varepsilon((\mu(x_{t,a_t}^T\hat{\theta}_{-1}) - r_t)x_{t,a_t})^T(\hat{\theta}_{-1} - \theta^\star)|\mathcal{F}_{t-1}] \\
=&\mathbb{E}[(\mu(x_{t,a_t}^T\hat{\theta}_{-1}) - \mu(x_{t,a_t}^T\theta^\star))x_{t,a_t}^T(\hat{\theta}_{-1} - \theta^\star)|\mathcal{F}_{t-1}] \\
\geq&\zeta\mathbb{E}[[x_{t,a_t}^T(\hat{\theta}_{-1} - \theta^\star)]^2|\mathcal{F}_{t-1}] \geq \zeta\kappa_l p_*/d\|\hat{\theta}_{-1} - \theta^\star\|^2,
\end{aligned}
$$

where the last inequality is from Lemma C.3 and Markov's inequality $\lambda_{\min}(\mathbb{E}_{x_{a_t}}[x_{a_t}x_{a_t}^T|\mathcal{F}_{t-1}]) \geq \kappa_l p_*/d$. Moreover, notice that $\|\hat{g}_t\| = r_{\varepsilon,\delta}$. Let $\lambda := 2\kappa_l\zeta p_*/d$ and $\eta_t = \frac{1}{\lambda t}$,

$$
\begin{aligned}
\|\hat{\theta}_t - \theta^\star\|^2 &= \|\hat{\theta}_{t-1} - \eta_t\hat{g}_t - \theta^\star\|^2 \\
&= \|\hat{\theta}_{t-1} - \theta^\star\|^2 - 2\eta_t\hat{g}_t^T(\hat{\theta}_{t-1} - \theta^\star) + \eta_t^2\|\hat{g}_t\|^2 \\
&= \|\hat{\theta}_{t-1} - \theta^\star\|^2 - 2\eta_t g_t^T(\hat{\theta}_{t-1} - \theta^\star) + 2\eta_t\hat{z}_t^T(\hat{\theta}_{t-1} - \theta^\star) + \eta_t^2\|\hat{g}_t\|^2 \\
&\leq (1 - 2\lambda\eta_t)\|\hat{\theta}_{t-1} - \theta^\star\|^2 + 2\eta_t\hat{z}_t^T(\hat{\theta}_{t-1} - \theta^\star) + \eta_t^2\|\hat{g}_t\|^2 \\
&\leq \left(1 - \frac{2}{t}\right)\|\hat{\theta}_{t-1} - \theta^\star\|^2 + \frac{2}{\lambda t}\hat{z}_t^{(}\hat{\theta}_{t-1} - \theta^\star) + \left(\frac{r_{\varepsilon,d}}{\lambda t}\right)^2.
\end{aligned}
$$

It follows from the same proof as in Proposition 1 in [28], we can obtain for any $0 < \alpha \leq \frac{1}{eT}, T \geq 4$ and for all $3 \leq t \leq T$, with probability at least $1 - \alpha$,

$$
\|\hat{\theta}_t - \theta^\star\|^2 \leq \frac{(624\log(\log(T)/\alpha) + 1)r_{\varepsilon,d}^2 d^2}{4\kappa_l^2\zeta^2 p_*^2 t}.
$$

$\square$

## C.3  Proof of Problem-dependent Bound

To prove the problem-dependent bound, we need only combine Lemma C.1 and Lemma C.2 together with the following lemma.

**Lemma C.4.** *Under the $(\beta, \gamma)$-margin condition, if we have $\|\hat{\theta}_t - \theta^\star\| \leq \dfrac{U_0}{\sqrt{t}}$ holds uniformly for all $t_0 \leq t \leq T_0$ for some $t_0$ and $U_0$ with probability at least $1 - \alpha$, we have then with probability at least $1 - 2\alpha$,*

$$
Reg(T) \leq C \cdot \left\{ \begin{array}{ll} c_r t_0 + \gamma(LC_B U_0)^2(\log T + o(1)), & \beta = 1 \\ c_r t_0 + \frac{2\gamma}{1-\beta}(LC_B U_0)^{1+\beta}(T^{\frac{1-\beta}{2}} + o(1)), & 0 \leq \beta < 1. \end{array} \right.
$$

*Proof.* We have, with probability at least $1 - \alpha$,

$$
\begin{aligned}
\text{Reg}(T) &\leq 2c_r t_0 + (\mu(x_{t,a_t^*}^T\theta^\star) - \mu(x_{t,a_t}^T\theta^\star))\mathbf{1}\{\|\hat{\theta}_t - \theta^\star\| \leq \frac{U_0}{\sqrt{t}}, \triangle_t \leq \frac{2LC_B U_0}{\sqrt{t}}\} \\
&\leq 2c_r t_0 + 2LC_B\frac{U_0}{\sqrt{t}}\mathbf{1}\{\triangle_t \leq \frac{2LC_B U_0}{\sqrt{t}}\}.
\end{aligned}
$$

Denote $A_t := \dfrac{1}{\sqrt{t}}\mathbf{1}\{\triangle_t \leq \dfrac{2LC_B U_0}{\sqrt{t}}\}$, by Hoeffding's inequality we have with probability at least $1 - \alpha$,

$$
\sum_t A_t < \sum_t \mathbb{E}[A_t] + \sqrt{\log T \log\frac{1}{\alpha}}.
$$

Noting that $\mathbb{E}[\sum_t A_t] \leq 2\gamma LC_B U_0 \log T$ for $\beta = 1$ and $\mathbb{E}[\sum_t A_t] \leq \dfrac{2\gamma}{1-\beta}(LC_B U_0)^\beta T^{\frac{1-\beta}{2}}$ for $0 \leq \beta < 1$. Then the claim holds. $\square$

## D  Proof of Results in Section 4.2

To lighten the notation, in this section we denote $\theta_i$ the underlying parameter of arm i. In the following analysis, without special explaination, all the $c$ and $C$ denote absolute constants. Sometimes we state the inequality of type $A_1 \leq C\log(A_2/\alpha)A_3$ holds with probability at least $1 - \alpha$ while in proof we derive the results hold with $1 - c\alpha$ for some constant c. In fact, they are equivalent by re-scaling $\alpha$ and changing $C$ to some larger constant.

### D.1 Proof of Theorem 4.1

**Lemma D.1.** *If after the warm up stage of length $Ks_0$, the estimator $\hat{\theta}_{Ks_0,i}$ achieves the following error bound with probability at least $1 - \alpha$,*

$$\sup_{i \in [K]} \|\hat{\theta}_{Ks_0,i} - \theta_i\| \leq h_0 := \frac{h_{sub}}{8LC_B},$$

*With $h = h_{sub}$ in Algorithm 2, we have $\mathbb{P}\{a_t^* \in \hat{K}_t, \hat{K}_t \cap K_{sub} = \emptyset\} \geq 1 - \alpha$ holds uniformly for all $Ks_0 < t \leq T$.*

*Proof.* Firstly, to show $a_t^* \in \hat{K}_t$, without loss of generality we assume that $a_t^* \neq 1$, and $\operatorname{argmax}_{i \in [K]} \mu(X_t^T \hat{\theta}_{Ks_0,i}) = 1$. Then by the optimality of $\theta_{a_t^*}$, condition on $\sup_{i \in [K]} \|\hat{\theta}_{Ks_0,i} - \theta_i\| \leq h_0$,

$$\mathbb{P}(a_t^* \notin \hat{K}_t) = \mathbb{P}(\mu(X_t^T \hat{\theta}_{Ks_0,a_t^*}) < \mu(X_t^T \hat{\theta}_{Ks_0,1}) - h/2)$$
$$\leq \mathbb{P}(\mu(X_t^T \theta_{a_t^*}) - h/8 < \mu(X_t^T \theta_1) + h/8 - h/2) = 0.$$

Now for any $j \in K_{sub}$, we have condition on $\sup_{i \in [K]} \|\hat{\theta}_{Ks_0,i} - \theta_i\| \leq h_0$,

$$\mathbb{P}(j \in \hat{K}_t) \leq \mathbb{P}(\mu(X_t^T \hat{\theta}_{Ks_0,a_t^*}) - h/2 < \mu(X_t^T \hat{\theta}_{Ks_0,j}))$$
$$\leq \mathbb{P}(\mu(X_t^T \theta_{a_t^*}) - 3h/4 < \mu(X_t^T \theta_j) + h/4) = 0,$$

where the final equation is due to the sub-optimality gap assumed in Assumption 5. □

*Proof of Theorem 4.1.* We first show the following lemma, which converts the regret bound under margin condition to the estimation error bound:

**Lemma D.2.** *Under the $(\beta, \gamma)$-margin condition, given $h_0$ defined in Lemma D.1, suppose there exists some $s_0$ such that with a warm up stage of length $Ks_0$, $\sup_{i \in [K]} \|\hat{\theta}_{t,i} - \theta_i\| \leq h_0$, and there exists some $t_0, U_0(\alpha)$ such that with probability at least $1 - \alpha$,*

$$\sup_{i \in K_{opt}} \|\hat{\theta}_{t,i} - \theta_i\| \leq \frac{U_0(\alpha)}{\sqrt{t}}, \quad \forall t_0 \leq t \leq T.$$

*Then, we have with probability at least $1 - 2\alpha$, for some constant $C$,*

$$Reg(T) \leq C \cdot \begin{cases} c_r t_0 + \gamma (LC_B U_0(\alpha))^2 (\log T + o(1)), & \beta = 1 \\ c_r t_0 + \dfrac{\gamma}{1 - \beta} (LC_B U_0(\alpha))^{1+\beta} (T^{\frac{1-\beta}{2}} + o(1)), & 0 < \beta < 1. \end{cases}$$

*Proof of Lemma D.2.* Denoting $E_t := \{\hat{K}_t \cap K_{sub} = \emptyset, a_t^* \in \hat{K}_t\}$, we have with probability at least $1 - \alpha$,

$$Reg(T) \leq 2c_r t_0 + L \sum_{t_0 < t \leq T} X_t^T (\theta_{a_t^*} - \theta_{a_t})$$

$$\leq 2c_r t_0 + L \sum_{t_0 < t \leq T} X_t^T (\theta_{a_t^*} - \theta_{a_t}) \mathbf{1}\{\sup_{i \in K_{opt}} \|\hat{\theta}_{t,i} - \theta_i\| \leq \frac{U_0(\alpha)}{\sqrt{t}}, E_t\}$$

$$\leq 2c_r t_0 + L \sum_{t_0 < t \leq T} X_t^T (\theta_{a_t^*} - \theta_{a_t}) \mathbf{1}\{\sup_{i \in K_{opt}} \|\hat{\theta}_{t,i} - \theta_i\| \leq \frac{U_0(\alpha)}{\sqrt{t}}, \triangle_t \leq \frac{2LC_B U_0(\alpha)}{\sqrt{t}}, E_t\}$$

$$\leq 2c_r t_0 + L \sum_{t_0 < t \leq T} \frac{2C_B U_0(\alpha)}{\sqrt{t}} \mathbf{1}\{\triangle_t \leq \frac{2LC_B U_0(\alpha)}{\sqrt{t}}\}.$$

Let $A_t = \mathbf{1}\{\triangle_t < \dfrac{2LC_B U_0(\alpha)}{\sqrt{t}}\}$. Then $A_t$ is a sequence of independent 0-1 valued random variable such that $\mathbb{P}(A_t = 1) \leq \gamma(\dfrac{2LC_B U_0(\alpha)}{\sqrt{t}})^\beta$. Then Hoeffding's inequality implies with probability at

least $1 - \alpha$,

$$\sum_{t_0 \leq t \leq T} \frac{1}{\sqrt{t}} A_t \leq \mathbb{E}[\sum_{1 \leq t \leq T} \frac{1}{\sqrt{t}} A_t] + \sqrt{\log T \cdot \log(\frac{1}{\alpha})}.$$

Notice that $\mathbb{E}[\sum_{1 \leq t \leq T} \frac{1}{\sqrt{t}} A_t] \leq CLC_B \gamma U_0(\alpha) \log T$ when $\beta = 1$ and $\mathbb{E}[\sum_{1 \leq t \leq T} \frac{1}{\sqrt{t}} A_t] \leq C \frac{\gamma}{1-\beta} (LC_B U_0(\alpha))^\beta T^{\frac{1-\beta}{2}}$ when $0 < \beta < 1$. This completes the proof. $\qquad \square$

Given Lemma D.2, we need only show that for both the private OLS estimator and the private SGD estimator, we can find the corresponding $s_0, t_0$ and $U_0(\alpha)$.

**Lemma D.3** (Result of OLS estimator). *Given* $h_0 = \dfrac{h_{sub}}{8LC_B}$ *and* $\lambda_0 = (2LC_B)^{-1}(\dfrac{p'}{2\gamma})^{1/\beta}$, $r_{opt} := |K_{opt}|/K$, *under the* $(\beta, \gamma)$-*margin condition ,*

$$s_0 = CK(\frac{C_B \sigma_\epsilon + \sigma_{\varepsilon,\delta}}{\min\{\lambda_0, h_0\} p' \kappa_l r_{opt}})^2 (d + \log(TK/\alpha)),$$
$$t_0 = 2K s_0,$$
$$U_0(\alpha) = \frac{K(C_B \sigma_\epsilon + \sigma_{\varepsilon,\delta})\sqrt{d + \log(TK/\alpha)}}{\kappa_l p'}.$$

*satisfy the requirements in Lemma D.2.*

**Lemma D.4** (Result of SGD estimator). *Given* $h_0 = \dfrac{h_{sub}}{8LC_B}$ *and* $\lambda_0 = (2LC_B)^{-1}(\dfrac{p'}{2\gamma})^{1/\beta}$, $r_{opt} := |K_{opt}|/K$, *under the* $(\beta, \gamma)$-*margin condition,*

$$s_0 = C \left( \frac{K r_{\varepsilon,d}}{\zeta \kappa_l p' r_{opt} \min\{\lambda_0, h_0\}} \right)^2 \log(KT \log(KT)/\alpha),$$
$$t_0 = K s_0 + 1,$$
$$U_0(\alpha) = C \frac{K\sqrt{\log((KT \log KT)/\alpha)} r_{\varepsilon,d}}{\zeta \kappa_l p'},$$

*satisfy the requirements in Lemma D.2.*

Then Theorem 4.1 follows from combining Lemma D.2, D.3 and D.4 . $\qquad \square$

**Remark.** *Notice that in the statement of Lemma D.3 and Lemma D.4, there exists a term* $r_{opt}$. *That is because of our assumption* $\mathbb{P}((v^T X)^2 \mathbf{1}\{X_t \in U_i\} > \kappa_l/K) > p'$. *In fact, a more natural assumption should be* $\mathbb{P}((v^T X)^2 \mathbf{1}\{X_t \in U_i\} > \kappa_l/|K_{opt}|) > p'$. *In that case, we have* $r_{opt} = 1$, *which leads to more refined results.*

The proof of Lemma D.3 and Lemma D.4 needs the following result: For a fixed $\beta \in (0,1]$, we define $h_0 = \dfrac{h_{sub}}{8LC_B}, \lambda_0 = (2LC_B)^{-1}(\dfrac{p'}{2\gamma})^{1/\beta}$, $A_t := \{\sup_{i \in K_{opt}} \|\hat{\theta}_{t,i} - \theta_i\| \leq \lambda_0\}, H_0 := \{\sup_{i \in [K]} \|\hat{\theta}_{Ks_0,i} - \theta_i\| \leq h_0\}$.

**Lemma D.5.** *Define* $\mathcal{F}_t$ *the filtration generated by* $\{X_i\}_{i \in [t]}, \{\epsilon_i\}_{i \in [t]}$ *together with all randomness from* $\{\psi_i\}_{i \in [t]}$. *Then we have:*

$$\lambda_{\min}(\mathbb{E}[X_t X_t \mathbf{1}\{a_t = i\}|\mathcal{F}_{t-1}]) \geq \frac{p' \kappa_l}{2K} \mathbf{1}_{A_{t-1}} \mathbf{1}_{H_0}, \quad \forall i \in K_{opt}.$$

*Proof.* We have for every unit vector $v$

$$\mathbb{E}[v^T X_t X_t^T v \mathbf{1}\{a_t = i\}|\mathcal{F}_{t-1}]$$

$$\geq \mathbf{1}_{H_0} \frac{\kappa_l}{K} \mathbb{E}[\mathbf{1}\{|v^T X \mathbf{1}\{X_t \in U_i\}|^2 \geq \kappa_l/K, a_t = i, A_{t-1}\}|\mathcal{F}_{t-1}]$$

$$\geq \mathbf{1}_{H_0} \mathbf{1}_{A_{t-1}} \frac{\kappa_l}{K} \mathbb{E}[\mathbf{1}\{|v^T X \mathbf{1}\{X_t \in U_i\}|^2 \geq \kappa_l/K\} - \mathbf{1}\{a_t \neq i, X_t \in U_i, A_{t-1}\}|\mathcal{F}_{t-1}]$$

$$\geq \mathbf{1}_{H_0} \mathbf{1}_{A_{t-1}} \frac{\kappa_l}{K} [p' - \mathbb{P}(\{a_t \neq i, X_t \in U_i\} \cap H_0 \cap A_{t-1}|\mathcal{F}_{t-1})].$$

$$\mathbb{P}(\{a_t \neq i, X_t \in U_i\} \cap H_0 \cap A_{t-1}|\mathcal{F}_{t-1}) = \mathbf{1}_{H_0} \mathbf{1}_{A_{t-1}} \mathbb{P}(\{a_t \neq i, X_t \in U_i\} \cap E_t \cap A_{t-1}|\mathcal{F}_{t-1})$$

$$\leq \mathbf{1}_{A_{t-1}} \mathbf{1}_{H_0} \mathbb{P}(\triangle_t < 2LC_B\lambda_0)$$

$$\leq \mathbf{1}_{A_{t-1}} \mathbf{1}_{H_0} \gamma (2LC_B\lambda_0)^\beta$$

$$\leq \mathbf{1}_{A_{t-1}} \mathbf{1}_{H_0} \frac{p'}{2},$$

where the last inequality is by the choice of $\lambda_0$. Then the proof is finished. $\square$

## D.2 Proof of Lemma D.3

We first establish the lower bound of the sample-covariance matrix sampled by the greedy action based on the following matrix-martingale concentration result:

**Lemma D.6** (Theorem 3.1 in [40])**.** *Let $z^1, \ldots, z^t$ be a sequence of random, positive-semidefinite $d \times d$ matrices adapted to a filtration $\mathcal{F}'_t$, let $Z_t := \sum_{i=1}^t z^i$ and $\tilde{Z}_t := \sum_{i=1}^t \mathbb{E}[z^i|\mathcal{F}'_{i-1}]$. Suppose that $\lambda_{\max}(z^i) \leq R^2$ almost surely for all $i$, then for any $\mu$ and $\alpha \in (0, 1)$,*

$$\mathbb{P}[\lambda_{\min}(Z_t) \leq (1 - \alpha)\mu, \lambda_{\min}(\tilde{Z}_t) \geq \mu] \leq d(\frac{1}{e^\alpha (1 - \alpha)^{1-\alpha}})^{\mu/R^2}.$$

Now we can show the following result:

**Lemma D.7.** *For $t_1 < t_2 \in \mathbb{N}$ such that $(t_2 - t_1) \cdot \frac{\kappa_l p'}{8K} > 10C_B^2 \log(d/\alpha')$, for a fixed $i \in [K]$ we have*

$$\mathbb{P}(\lambda_{\min}(\sum_{t=t_1}^{t_2} X_t X_t \mathbf{1}\{a_t = i\}) \leq \frac{t_2 - t_1}{8K} \kappa_l p', \sup_{t_1 \leq t \leq t_2, i \in K_{opt}} \|\hat{\theta}_{t,i} - \theta_i\| \leq \lambda_0, H_0) \leq \alpha'.$$

*Proof.* Denote $S_{t_1, t_2} := \cap_{t_1 \leq t \leq t_2} A_t$, by Lemma D.5 we have

$$\lambda_{\min}(\sum_{t=t_1}^{t_2} \mathbb{E}[X_t X_t^T \mathbf{1}\{a_t = i\}|\mathcal{F}_{t-1}]) \geq \sum_{t=t_1}^{t_2} \mathbf{1}_{A_{t-1}} \mathbf{1}_{H_0} \frac{\kappa_l p'}{2K}.$$

That implies

$$\mathbb{P}(\lambda_{\min}(\sum_{t=t_1}^{t_2} X_t X_t^T \mathbf{1}\{a_t = i\}) \leq \frac{t_1 - t_2}{4K} \kappa_l p', S_{t_1, t_2}, H_0)$$

$$\leq \mathbb{P}(\lambda_{\min}(\sum_{t=t_1}^{t_2} X_t X_t^T \mathbf{1}\{a_t = i\}) \leq \frac{t_1 - t_2}{4K} \kappa_l p', \mathbb{E}[X_t X_t^T \mathbf{1}\{a_t = i\}|\mathcal{F}_{t-1}]) \geq (t_2 - t_1) \frac{\kappa_l p'}{2K}).$$

Then selecting $\alpha = 1/2$ and $\mu = (t_2 - t_1) \cdot \frac{\kappa_l p'}{4K}$ in Lemma D.6, we have

$$\mathbb{P}(\lambda_{\min}(\sum_{t=t_1}^{t_2} X_t X_t^T \mathbf{1}\{a_t = i\}) \leq (t_2 - t_1) \frac{\kappa_l p'}{8K}, S_{t_1, t_2}, H_0) \leq d(\frac{1}{\sqrt{e/2}})^{10 \log(\frac{d}{\alpha'})} \leq \alpha'.$$

That leads to the claim. $\square$

In warm up stage, we have the following lemma.

**Lemma D.8.** *As long as $s_0 \geq C(r_{opt}\kappa_l p')^{-2}\max\{\log\frac{1}{\alpha}, d\}$ for some absolute constant C, we have with probability at least $1 - \alpha$,*

$$\lambda_{\min}(\sum_{t=1}^{Ks_0}\mathbf{1}\{a_t = i\}X_tX_t^T)^{-1} \leq \frac{2}{s_0p'r_{opt}\kappa_l}, \quad \forall i \in [K].$$

*Proof.* Since $X_t$ are i.i.d. for $(i-1)s_0 + 1 \leq t \leq is_0$, using classical concentration results for i.i.d. sub-gaussian covariance matrix result (e.g. Theorem 6.5 in [42] ), we have when $s_0 > C(r_{opt}\kappa_l p')^{-2}\max\{\log\frac{1}{\alpha}, d\}$, with probability at least $1 - \alpha$,

$$\|\frac{1}{s_0}\sum_{t=1}^{Ks_0}\mathbf{1}\{a_t = i\}X_tX_t^T - \mathbb{E}[X_1X_1^T]\| \leq c_1(\sqrt{\frac{d}{s_0}} + \frac{d}{s_0}) + c_2\max\{\sqrt{\frac{\log 1/\alpha}{s_0}}, \frac{\log 1/\alpha}{s_0}\}$$

$$\leq c_3(\sqrt{\frac{d}{s_0}} + \sqrt{\frac{\log(1/\alpha)}{s_0}})$$

$$\leq r_{opt}p'\kappa_l/2.$$

On the other hand, we have by Markov's inequality

$$\lambda_{\min}\mathbb{E}[X_1X_1^T] \geq \sum_{i \in K_{opt}}\lambda_{\min}\mathbb{E}[X_1X_1^T\mathbf{1}\{X_1 \in U_i\}] \geq r_{opt}\kappa_l p'.$$

Thus we have with probability at least $1 - \alpha$,

$$\lambda_{\min}(\sum_{t=1}^{Ks_0}X_tX_t^T) \geq s_0r_{opt}p'\kappa_l/2.$$

$\square$

Now we can claim our first result about the private OLS-estimator in the warm up stage:

**Lemma D.9.** *Selecting $s_0$ as in Lemma D.8 . For the warm up stage with private-OLS-estimator and length $Ks_0$, we have for any $\alpha > 0$, with probability at least $1 - \alpha$,*

$$\sup_{i \in [K]}\|\hat{\theta}_{t,i} - \theta_i\| \leq \frac{(4C_B\sigma_\epsilon + \sigma_{\varepsilon,\delta})\sqrt{t(\log(\frac{TK}{\alpha}) + d)}}{s_0p'r_{opt}\kappa_l} \quad \text{holds for all } Ks_0 \leq t \leq T.$$

*Proof.* Denote $U_t = \sum_{s=1}^{t}(\mathbf{1}\{a_s = i\}X_sX_s^T + (\mathbf{1}\{a_s = i, s \leq Ks_0\} + \mathbf{1}\{s > Ks_0\})W_s) + \tilde{c}\sqrt{t}I_d$, we have

$$\hat{\theta}_{t,i} = U_t^{-1}(\sum_{s=1}^{t}\mathbf{1}\{a_s = i\}X_sy_s + (\mathbf{1}\{a_s = i, s \leq Ks_0\} + \mathbf{1}\{s > Ks_0\})\xi_s)$$

$$= U_t^{-1}(\sum_{s=1}^{t}\mathbf{1}\{a_s = i\}[X_sX_s^T\theta_i + X_s\epsilon_s] + (\mathbf{1}\{a_s = i, s \leq Ks_0\} + \mathbf{1}\{s > Ks_0\})\xi_s)$$

$$= \theta_i + U_t^{-1}(\sum_{s=1}^{t}(\mathbf{1}\{a_s = i\}X_s\epsilon_s + (\mathbf{1}\{a_s = i, s \leq Ks_0\} + \mathbf{1}\{s > Ks_0\})(\xi_s - W_s\theta_i)) - \tilde{c}\sqrt{t}I_d\theta_i).$$

By $\|\sum_{s=1}^{Ks_0}\mathbf{1}\{a_s = i\}W_s + \sum_{s=Ks_0+1}^{t}W_s\| \leq \tilde{c}\sqrt{t}, \forall Ks_0 \leq t \leq T, i \in [K]$ with probability at least $1 - \alpha$, we have with probability at least $1 - 2\alpha$,

$$[\lambda_{\min}(U)]^{-1} \leq \lambda_{\min}(\sum_{s=1}^{Ks_0}\mathbf{1}\{a_s = i\}X_sX_s^T)^{-1} \leq \frac{2}{s_0r_{opt}p'\kappa_l}, \quad \forall Ks_0 \leq t \leq T.$$

On the other hand, we have by the concentration of sub-gaussian random vector, the following bounds hold with probability at least $1 - \alpha/(T^2 K)$:

$$\|\sum_{s=1}^{t} \mathbf{1}\{a_s = i\} X_s \epsilon_s\| \leq C C_B \sigma_\epsilon \sqrt{t(d + \log(TK/\alpha))}, \tag{17}$$

$$\|\sum_{s=1}^{t} (\mathbf{1}\{a_s = i, s \leq K s_0\} + \mathbf{1}\{s > K s_0\})\xi_s\| \leq C \sigma_{\varepsilon,\delta} \sqrt{t(d + \log(TK/\alpha))}, \tag{18}$$

$$\|\sum_{s=1}^{K s_0} \mathbf{1}\{a_s = i\} W_s \theta_i + \sum_{s=K s_0 + 1}^{t} W_s \theta_i\| \leq \tilde{c}\sqrt{t}\|\theta_i\| \leq C \sigma_{\varepsilon,\delta} \sqrt{t(d + \log(TK/\alpha))}. \tag{19}$$

Gathering all bounds together, we have with probability at least $1 - (2 + \frac{1}{T^2})\alpha$,

$$\sup_{i \in [K]} \|\hat{\theta}_{t,i} - \theta_i\| \leq \frac{2C}{s_0 p' r_{opt} \kappa_l}(C_B \sigma_\epsilon + \sigma_{\varepsilon,\delta}) \sqrt{t(\log(TK/\alpha) + d)}.$$

That finishes the proof. $\qquad\qquad\square$

**Lemma D.10.** *As long as*

$$s_0 \geq C K \left(\frac{C_B \sigma_\epsilon + \sigma_{\varepsilon,\delta}}{\min\{\lambda_0, h_0\} r_{opt} p' \kappa_l}\right)^2 (d + \log(TK/\alpha)),$$

*we have with probability at least $1 - \alpha$,*

$$\sup_{i \in [K]} \|\hat{\theta}_{K s_0,i} - \theta_i\|_2 \leq \min\{\lambda_0, h_0\}, \tag{20}$$

$$\sup_{i \in K_{opt}} \|\hat{\theta}_{s,i} - \theta_i\|_2 \leq \lambda_0 \text{ holds uniformly for } K s_0 \leq s \leq (K+1)s_0, \tag{21}$$

$$C\frac{K(C_B \sigma_\epsilon + \sigma_{\varepsilon,\delta})\sqrt{d + \log(TK/\alpha)}}{\sqrt{t - K s_0}\kappa_l p'} \leq \lambda_0 \text{ holds for all } t \geq 2K s_0 . \tag{22}$$

*Proof.* To show (20),(21), we can just plug the value of $s_0$ into the upper bound in Lemma D.9. (22) comes directly from the value of $s_0$. $\qquad\qquad\square$

Now, we can show the following result:

**Lemma D.11.** *With the choice of $s_0$ same as in Lemma D.10, for $t > K s_0$, denote $t' = t - K s_0$ and $\tilde{t}_0 = 2K s_0$, we have if*

$H_0$ *holds and* $\|\hat{\theta}_{t,i} - \theta_i\|_2 \leq \min\{\tilde{U}_s(\alpha), \lambda_0\}$ *holds uniformly for* $i \in K_{opt}, \tilde{t}_0 \leq s \leq t$ ,

*with probability at least* $1 - \sum_{j=1}^{t'} \frac{2}{j^2}\alpha$*, then*

$H_0$ *holds and* $\|\hat{\theta}_{t,i} - \theta_i\|_2 \leq \min\{\tilde{U}_s(\alpha), \lambda_0\}$ *holds uniformly for* $i \in K_{opt}, \tilde{t}_0 \leq s \leq t+1$ ,

*with probability at least* $1 - \sum_{j=1}^{t'+1} \frac{2}{j^2}\alpha$ *, where*

$$\tilde{U}_s(\alpha) = C\frac{K(C_B \sigma_\epsilon + \sigma_{\varepsilon,\delta})\sqrt{d + \log(TK/\alpha)}}{\sqrt{s}\kappa_l p'}.$$

*Proof.* Denote $S_{\tilde{t}_0,t} = \{\|\hat{\theta}_{s,i} - \theta_i\| \leq \min\{\tilde{U}_s(\alpha), \lambda_0\}, \forall K \in K_{opt}, \forall \tilde{t}_0 \leq s \leq t\}, \tilde{A}_t = \{\sup_{i \in K_{opt}} \|\hat{\theta}_{i,t} - \theta_i\| \leq \tilde{U}_t(\alpha)\}$ , we have by Lemma D.7

$$\mathbb{P}(S_{\tilde{t}_0,t}, H_0, \lambda_{\min}(\sum_{s=1}^{t} X_s X_s \mathbf{1}\{a_s = i\}) > \frac{t'\kappa_l p'}{8K}) \geq 1 - \frac{\alpha}{2KT^2}.$$

Applying the inequalities (17),(18) (19), we have

$$\mathbb{P}(H_0, S_{\tilde{t}_0,t}, A_{t+1}) \geq 1 - \sum_{j=1}^{t'} \frac{2}{j^2}\alpha - \frac{3\alpha}{2T^2} - \sum_{i \in K_{opt}} \mathbb{P}(H_0, S_{\tilde{t}_0,t}, \tilde{A}_{t+1}, \lambda_{\min}(\sum_{s=1}^{t} X_s X_s \mathbf{1}\{a_s = i\}) \leq \frac{t'\kappa_l p'}{4})$$

$$\geq 1 - \sum_{j=1}^{t'} \frac{2}{j^2}\alpha - \frac{2\alpha}{T^2}$$

$$\geq 1 - 2\sum_{j=1}^{t'+1} \frac{1}{j^2}\alpha.$$

By the selection of $s_0$, we have $\tilde{U}_s(\alpha) \leq \lambda_0$ for $\tilde{t}_0 \leq s \leq t+1$, and as a result, $\mathbb{P}(H_0, S_{\tilde{t}_0,t+1}) = \mathbb{P}(H_0, S_{\tilde{t}_0,t}, \tilde{A}_{t+1})$. Thus the claim holds. $\qquad\square$

*Proof of Lemma D.3.* Lemma D.3 is implied directly by Lemma D.11 and Lemma D.10.

$\square$

### D.3   Proof of Lemma D.4

*Proof.* For the estimator $\hat{\theta}_{Ks_0,i}$ at the end of warm up stage, since the action is independent of the contexts, every $\hat{\theta}_{Ks_0,i}$ can be seen as an output of performing private gradient descent over $s_0$ i.i.d. samples. Without loss of generality, we perform the analysis for the parameter of the first arm $\hat{\theta}_{Ks_0,1}$ (notice that by the sampling strategy in the warm up stage, we have $\hat{\theta}_{Ks_0,1} = \hat{\theta}_{s_0,1}$). The result for other $\hat{\theta}_{Ks_0,i}$ can be established using the same argument. For $2 \leq t \leq s_0$,

$$\|\hat{\theta}_{t,i} - \theta_i\|^2 = \|\hat{\theta}_{t-1,i} - \eta_t \hat{g}_t - \theta_i\|^2$$
$$= \|\hat{\theta}_{t-1,i} - \theta_i\|^2 - 2\eta_t \hat{g}_t^T(\hat{\theta}_{t-1,i} - \theta_i) + 2\eta_t^2 \|\hat{g}_t\|^2$$

Here $\hat{g}_t := \Psi_\varepsilon[(\mu(X_t^T \hat{\theta}_{t,i}) - r_t)X_t]$, by the unbiasedness of $\Psi_\varepsilon$ in Lemma 2.2 we have

$$\mathbb{E}[\Psi_\varepsilon((\mu(X_t^T \hat{\theta}_{t-1,i}) - r_t)X_t)^T(\hat{\theta}_{t-1,i} - \theta_i)|\mathcal{F}_{t-1}]$$
$$= \mathbb{E}[(\mu(X_t^T \hat{\theta}_{t-1,i}) - \mu(X_t^T \theta_i))X_t^T(\hat{\theta}_{t-1,i} - \theta_i)|\mathcal{F}_{t-1}]$$
$$\geq \zeta\mathbb{E}[[X_t^T(\hat{\theta}_{t-1,i} - \theta_i)]^2|\mathcal{F}_{t-1}]$$
$$\geq \zeta\kappa_l r_{opt} p' \|\hat{\theta}_{t-1,i} - \theta_i\|^2.$$

We get

$$\|\hat{\theta}_{t,i} - \theta_i\|^2 \leq (1 - 2\zeta r_{opt}\kappa_l p'\eta_t)\|\hat{\theta}_{t-1,i} - \theta_i\|^2 + 2\eta_t(\mathbb{E}[\hat{g}_t|\mathcal{F}_{t-1}] - \hat{g}_t)^T(\hat{\theta}_{t-1,i} - \theta_i) + 2\eta_t^2\|\hat{g}_t\|^2.$$

Notice $\|\hat{g}_t\|_2^2$ is upper bounded by $r_{\varepsilon,d}^2$. Now using the same argument as in the proof of Proposition 1 of [28] leads to the following result:

**Lemma D.12.** *If we pick $\eta_t = 1/(r_{opt}\zeta\kappa_l p't)$ in the warm up stage, then with probability at least $1 - \alpha$,*

$$\sup_{i \in [K]} \|\hat{\theta}_{Ks_0,i} - \theta_i\|^2 \leq C\frac{(\log(\log(KT)/\delta) + 1)r_{\varepsilon,\delta}^2}{\zeta^2\kappa_l^2 r_{opt}^2 p'^2 s_0}. \tag{23}$$

Notice that in our algorithm, when $t > Ks_0$, for any $i \in K_{opt}$, the private gradient descent formula is given by

$$\hat{\theta}_{t,i} = \hat{\theta}_{t-1,i} - \eta_t \tilde{g}_t,$$

with $\tilde{g}_t = \mathbf{1}\{a_t = i\}\hat{g}_t + \mathbf{1}\{a_t \neq i\}\Psi_\varepsilon(0)$. Again without loss of generality we assume that $1 \in K_{opt}$, and we provide the analysis for $i = 1$, the argument is same for other $i \in K_{opt}$:

$$
\begin{aligned}
\mathbb{E}[\tilde{g}^T(\hat{\theta}_{t-1,1} - \theta_1)|\mathcal{F}_{t-1}] &= \mathbb{E}[\mathbf{1}\{a_t = i\}\hat{g}^T(\hat{\theta}_{t-1,1} - \theta_1)|\mathcal{F}_{t-1}] \\
&= \mathbb{E}[\mathbf{1}\{a_t = i\}(\mu(X_t^T\hat{\theta}_{t-1,1}) - \mu(X_t^T\theta_i))X_t^T(\hat{\theta}_{t-1,1} - \theta_1)|\mathcal{F}_{t-1}] \\
&\geq \zeta\mathbb{E}[\mathbf{1}\{a_t = i\}[X_t^T(\hat{\theta}_{t-1,1} - \theta_1)]^2|\mathcal{F}_{t-1}] \\
&\geq \mathbf{1}_{A_t,H_0}\zeta\kappa_l p'\eta_t\|\hat{\theta}_{t-1,1} - \theta_1\|^2/K
\end{aligned}
$$

select $\eta_t := K/(\zeta\kappa_l p't')$, with $t' = t - (K-1)s_0$ we have then

$$
\|\hat{\theta}_{t,1} - \theta_1\|_2^2 \leq (1 - \frac{2}{t'}\mathbf{1}_{A_t,H_0})\|\hat{\theta}_{t-1,1} - \theta_1\|^2 + \frac{2K}{\zeta\kappa_l p't'}(\mathbb{E}[\tilde{g}_t|\mathcal{F}_{t-1}] - \tilde{g}_t)^T(\hat{\theta}_{t-1,1} - \theta_1) + 2(\frac{Kr_{\varepsilon,d}}{\zeta\kappa_l p't'})^2
$$

If we denote $S_t := \cap_{s=Ks_0}^t A_s$, then using the above inequality recursively until $t = Ks_0 + 1$(i.e. until $t' = s_0 + 1$), we have

$$
\begin{aligned}
\mathbf{1}_{S_{t-1},H_0}\|\hat{\theta}_{t,1} - \theta_1\|^2 &\leq \frac{s_0(s_0-1)}{t'(t'-1)}\|\hat{\theta}_{Ks_0,1} - \theta_1\|^2 + 2(\frac{Kr_{\varepsilon,d}}{\zeta\kappa_l p't'})^2 \\
&\quad + \frac{2K}{(t'-1)t'\zeta\kappa_l p'}\sum_{s=Ks_0+1}^t (\mathbb{E}[\tilde{g}_s|\mathcal{F}_{t-1}] - \tilde{g}_s)^T(\hat{\theta}_{s-1,1} - \theta_1).
\end{aligned}
$$

Then it follows from the same proof as in Proposition 1 in [28] that for any fixed $Ks_0 < t \leq T$, we have with probability at least $1 - \alpha/T$,

$$
\mathbf{1}_{S_{t-1},H_0}\|\hat{\theta}_{t,1} - \theta_1\|^2 \leq \frac{s_0(s_0-1)}{t'(t'-1)}\|\hat{\theta}_{Ks_0,1} - \theta_1\|^2 + C\frac{K^2(\log(TK\log(TK)/\alpha) + 1)r_{\varepsilon,d}^2}{\zeta^2\kappa_l^2 p'^2 t'},
\tag{24}
$$

Now choose $s_0 \geq 2C\frac{K^2(\log(TK\log(TK)/\alpha) + 1)r_{\varepsilon,d}^2}{r_{opt}^2\zeta^2\kappa_l^2 p'^2\min\{\lambda_0, h_0\}^2}$, so that the second term in (24) is less or equal to $\lambda_0/2$, we have $\mathbb{P}(S_{Ks_0+1}, H_0) \geq 1 - 2\alpha$ by (23). And by calling (24) recursively we can get $\mathbb{P}(S_{t-1}, H_0) > 1 - 2\alpha - \frac{t - Ks_0}{T}\alpha \geq 1 - 3\alpha, \forall Ks_0 < t \leq T$. Then with probability at least $1 - 3\alpha$, we have

$$
\|\hat{\theta}_{t,1} - \theta_1\|^2 \leq C\frac{K^2(\log(3TK\log(TK)/\alpha))r_{\varepsilon,d}^2}{\zeta^2\kappa_l^2 p'^2(t - (K-1)s_0)}, \quad \forall Ks_0 < t \leq T.
$$

The above inequality is because the term $\frac{s_0(s_0-1)}{t'(t'-1)}\|\hat{\theta}_{Ks_0,1} - \theta_1\|^2 \leq \frac{s_0(s_0-1)}{t'(t'-1)}\frac{\min\{\lambda_0, h_0\}}{2}$, which can be absorbed into the constant $C$. $\qquad\square$

# E   Proof of Theorem 3.2

In this section, we would give a proof on the Theorem 3.2 by combining the argument in [22] and the divergence contraction inequality in [15].

*Proof of Theorem 3.2.* Consider the two-arm stochastic contextual bandit environment: for each d-dimensional context $i = 1$ or 2, $x_{t,i} \sim \mathcal{N}(0, \frac{1}{d}I_d)$ independently. If choosing action $a_t$ at time t, the reward $y_t$ is generated via $y_t = x_{t,a_t}^T\theta + \epsilon_t$ with $\epsilon_t \sim_{i.i.d.} \mathcal{N}(0,1)$. Given any fixed $\varepsilon$-LDP bandit algorithm $\pi$ with $\varepsilon \leq 1$, we denote its decision at $t$-th step by $a_t$, by definition $a_t$ can be seen as a function of current contextual $x_{t,1}, x_{t,2}$ and all history outputs $(x_{1,a_1}, y_1, x_{2,a_2}, y_2, \ldots, x_{t-1,a_{t-1}}, y_{t-1})$. Since the algorithm is under the $\varepsilon$-LDP constraint, each $a_t$ can only access $S_t := (M_1(x_{1,a_1}, y_1), M_2(x_{2,a_2}, y_2), \ldots, M_{t-1}(x_{t-1,a_{t-1}}, y_{t-1}))$ with $M_1, \ldots, M_{t-1}$ a sequence of $\varepsilon$-LDP mechanisms. We denote the distribution of $S_t$ by $Q_\theta^t$, and we have

$$
\begin{aligned}
&\mathbb{E}_{\theta \sim Q_0}[\mathbb{E}_{Q_\theta}^t[(x_{t,a_t^*} - x_{t,a_t})^T\theta|x_{t,1}, x_{t,2}]] \\
&= \mathbb{E}_{\theta \sim Q_0}[((x_{t,1} - x_{t,2})^T\theta)_+Q_\theta^t(a_t(S_t, x_t) = 2) + ((x_{t,2} - x_{t,1})^T\theta)_+Q_\theta^t(a_t(S_t, x_t) = 1)],
\end{aligned}
\tag{25}
$$

where $(x)_+$ denote $\max\{x, 0\}$ and $Q_0$ denote the uniform distribution over $\triangle S_1^{d-1}$ with $\triangle > 0$ some positive number to be determined, we define $Q_1, Q_2$ as

$$\frac{dQ_1}{dQ_0} := \frac{((x_{t,1} - x_{t,2})^T \theta)_+}{Z_0}, \quad \frac{dQ_2}{dQ_0} := \frac{((x_{t,2} - x_{t,1})^T \theta)_+}{Z_0},$$

where $Z_0 = \mathbb{E}_{Q_0}[((x_{t,1} - x_{t,2})^T \theta)_+] = \mathbb{E}_{Q_0}[((x_{t,2} - x_{t,1})^T \theta)_+]$ is the normalization factor. Denote $r_t = \|x_{t,1} - x_{t,2}\|, u_t = r_t^{-1}(x_{t,1} - x_{t,2})$, then the right hand side of (25) is lower bounded by

$$
\begin{aligned}
&= Z_0(Q_1 \circ Q_\theta^t(a_t(S_t, x_t) = 2) + Q_2 \circ Q_\theta^t(a_t(S_t, x_t) = 1)) \\
&\geq_{(a)} Z_0(1 - \mathrm{TV}(Q_1 \circ Q_\theta^t, Q_2 \circ Q_\theta^t)) \\
&\geq_{(b)} \frac{Z_0}{2} \exp(-D_{KL}(Q_1 \circ Q_\theta^t \| Q_2 \circ Q_\theta^t)) \\
&=_{(c)} \frac{Z_0}{2} \exp(-D_{KL}(Q_1 \circ Q_\theta^t \| Q_1 \circ Q_{\theta - 2(u_t^T \theta) u_t}^t)) \\
&\geq_{(d)} \frac{Z_0}{2} \exp(-\mathbb{E}_{Q_1}[D_{KL}(Q_\theta^t \| Q_{\theta - 2(u_t^T \theta) u_t}^t)]),
\end{aligned}
\tag{F.1}
$$

where $D_{KL}(\cdot \| \cdot)$ denote the KL-divergence, $TV(\cdot, \cdot)$ denote the total variation distance and $Q_i \circ Q_\theta^t$ means $\mathbb{E}_{\theta \sim Q_i}[Q_\theta^t]$. The (a) inequality comes from the fundamental limit of two-point testing (see e.g. Section 15.2 in [42]), and the (b) inequality comes from Lemma 2.6 of [41], the (c) equality comes from Lemma 8 in [22] and the (d) inequality comes from the strongly-convexity of KL-divergence. Now by chain rule of KL-divergence, the divergence contraction inequality in Theorem 1 of [15] and the formula of KL-divergence between Gaussian distributions, we have

$$
\begin{aligned}
D_{KL}(Q_\theta^t \| Q_{\theta - 2(u_t^T \theta) u_t}^t) &= \sum_{s=1}^{t-1} \mathbb{E}_{Q_\theta^{s-1}}[D_{KL}(P_\theta^t(\cdot | S_{s-1}) \| P_{\theta - 2(u_t^T \theta) u_t}^t(\cdot | S_{s-1}))] \\
&\leq \sum_{s=1}^{t-1} \frac{c}{2}(e^\varepsilon - 1)^2(2(u_t^T \theta)^2 \|u_t\|^2)
\end{aligned}
$$

By the argument of in [22], we have (F.1) is lower bounded by

$$\frac{r_t \triangle}{C\sqrt{d}} \exp(-C \frac{(e^\varepsilon - 1)^2 \triangle^2}{d+1} u_t^T (\sum_{s=1}^{t-1} x_{t,a_t} x_{t,a_t}^T) u_t).$$

Now taking expectation over $x_{t,1}, x_{t,2}$, and using the convexity of function $f(x) = \exp(-x)$ we get

$$\mathbb{E}_x \mathbb{E}_\theta \mathbb{E}_{Q_\theta^t}[x_{t,a_t^*} - x_{t,a_t}^T] \geq \frac{\triangle}{C\sqrt{d}} \exp(-\frac{C(e^\varepsilon - 1)^2 \triangle^2 t}{d^2}).$$

Selecting $\triangle \asymp \frac{d}{(e^\varepsilon - 1)\sqrt{t}}$ and taking summation over $1 \leq t \leq T$ leads to $\Omega(\sqrt{Td}/(e^\varepsilon - 1))$ lower bound, finally noticing $e^\varepsilon - 1 \leq C\varepsilon$ for $\varepsilon \leq 1$ leads to the desired lower bound when $\varepsilon \leq 1$. $\qquad \square$

## F  Additional Experiments

In this section, We evaluate all the four methods on two different privacy levels $\varepsilon = 0.5$ and $1$ in a larger scale scheme. To be specific, for single-param setting we increase dimension $d$ to 20 and increase the number of arms $K$ to 20; for multi-param setting we increase dimension $d$ to 10 and increase the number of the arms $K$ to 10.

In this simulation we change the learning rate scheme of LDP-SGD from $\eta_t = c_1 d/(\kappa_l \zeta p_* t)$ to $\eta_t = c_2 d/(\kappa_l \zeta p_* \sqrt{t})$ for some $c_2 > 1$ for its better empirical performance, while other details in data generation process are the same as in Section 5. The first and second columns in Figure 3 are for single-param and multi-param settings, respectively, which are simulation studies on linear bandits. As we can see the proposed LDP-OLS and LDP-SGD algorithms can still achieve better performance against their competitors under different privacy constraints.

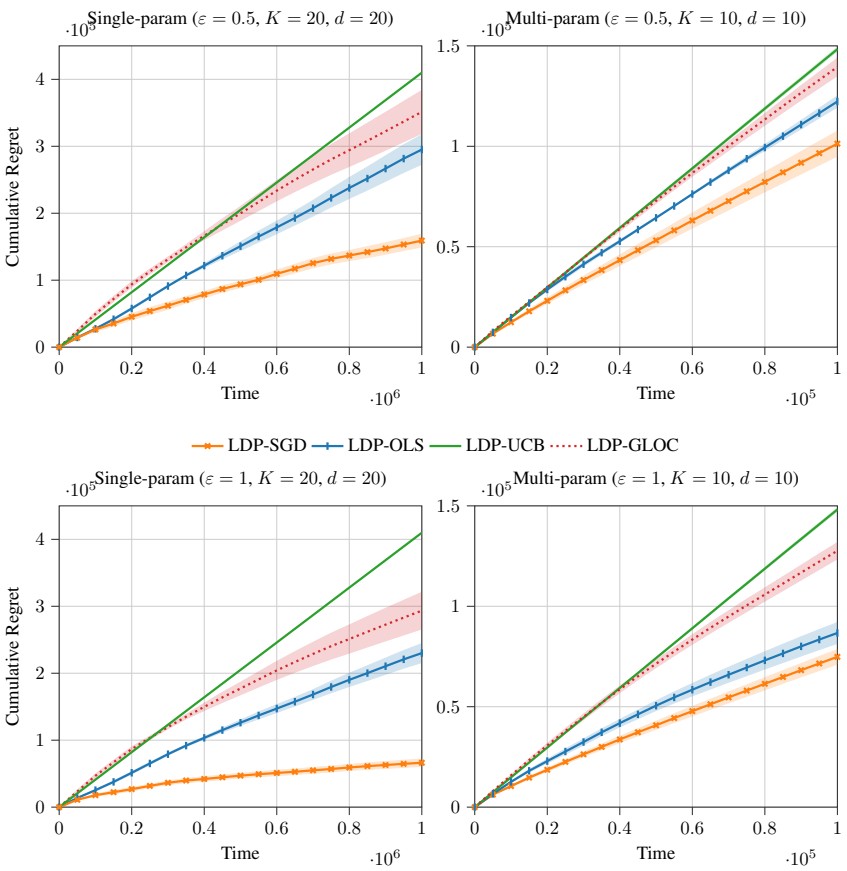

Figure 3: Simulation study in larger-scale scheme. We perform 10 replications for each case and plot the mean and 0.5 standard deviation of their regrets.

# G  Auto Loan Experiment Details

We use the same features selection as in [1, 11] in the dataset and select FICO score, the term of contract, the loan amount approved, prime rate, the type of car, and the competitor's rate as the feature vector for each customer. Note that a description of the data set (with descriptive statistics on the demand and available features) is available in [1]. The objective is to offer a personalized lending price (from a range of choices) based on personal information such as FICO score to a customer who will either accept or reject it. In contrast to linear bandits, the binary reward is non-linear. Therefore we leave LDP-UCB and LDP-OLS out of considerations. To formulate a bandit environment, first we need to recover the underlying true parameter. Since the lender's decision, i.e., the price for each customer, is not presented in the dataset, we follow [1, 11] and impute it by using the net-present value of futher payment minus the loan amount, i.e.,

$$p = \text{ Monthly Payment } \times \sum_{\tau=1}^{\text{Term}} (1 + \text{ Rate })^{-\tau} - \text{ Loan Amount }.$$

After imputing the loan prices, to represent customers' binary loan choices, we employ the logit demand model. To be specific, given a price $p$ and a context $x \in \mathbb{R}^d$, the binary variable *apply* takes value of 1 with probability $\frac{\exp(v)}{1+\exp(v)}$ and takes value of 0 with probability $\frac{1}{1+\exp(v)}$ where the linear predictor $v = (x, px)^T \theta^\star$. We conduct one-hot encoding for categorical features in the dataset and use the python package `sklearn` [25] for the estimation of the underlying parameter $\theta^\star$. We use the interval $[0, 25000]$ as the feasible region of the prices, which covers the lending prices computed from the dataset, and we discrete the feasible region uniformly into 25 options $\{p_i\}_{i \in [25]}$. We use LDP-SGD and LDP-GLOC to sequentially compute the loan prices for the 100,000 with

randomly selected customers in the dataset , and compute the company's expected regret based on the population model mentioned above under two privacy constraints scheme $\varepsilon = 0.5$ and $\varepsilon = 1$.

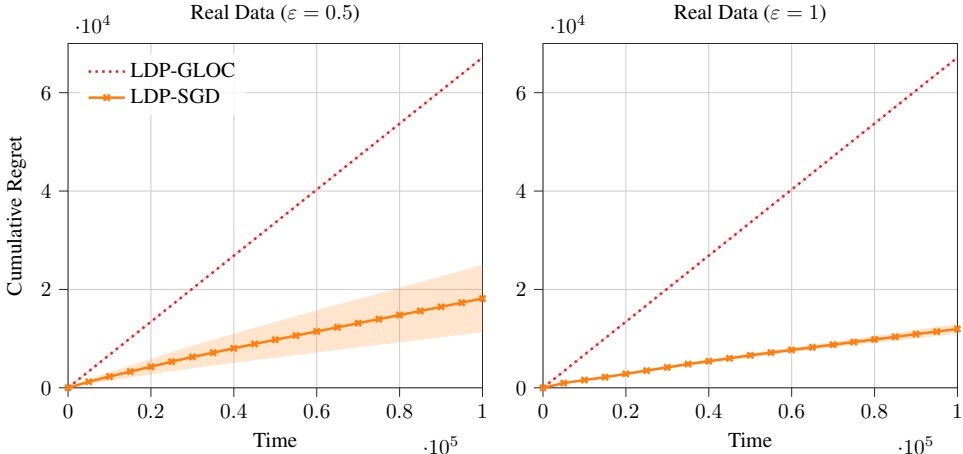

Figure 4: We perform 10 replications for each case and plot the mean and 0.5 standard deviation of their regrets.