# OpenReview forum: "Generalized Linear Bandits with Local Differential Privacy "
_NeurIPS.cc/2021/Conference — NeurIPS 2021 Poster_

### Official Review · Reviewer_SZ1X · 2021-07-12

**Rating:** 7
**Confidence:** 2

**Summary:**

The paper studies (generalized) linear contextual bandits in the context of local differential privacy (DP). At every time step a single user’s sensitive data point arrives, which serves as a context, and a reward is sampled as a function of this context and an unknown (generalized) linear reward-generating function. The difficulty with minimizing regret with local DP is that one has to learn the reward-generating function without compromising users’ privacy. The analysis is split by cases: in the setting of linear rewards, an OLS-based solution is proposed, and in the setting of generalized linear rewards, an SGD-based solution is proposed. In addition, two different bandit models are considered depending on whether the rewards from different arms are related.

**Limitations And Societal Impact:**

The paper discusses several limitations (as natural directions for future work). It doesn't discuss potential negative societal impacts, but the work is largely theoretical and builds on well-studied models in the literature, so I do not see potential societal concerns.

**Main Review:**

I am not very familiar with the works most closely related to this paper (e.g. [36]), but it appears that this paper is the first one to combine (1) local DP (2) stochastic generalized linear bandits. Prior work has studied joint DP + generalized linear bandits [8] or local DP + generalized linear bandits [36], but in an adversarial setting. So in the context of recent work the problem studied in this submission arises naturally.

 I am not sure if there is a significant complication in the linear case, where an OLS-based solution is proposed. In particular, the solution is based on perturbing the sufficient statistics, similar to, e.g., [29] and [36]. What is the main technical difference here?
 The SGD solution is elegant and arguably more interesting. Even without privacy concerns, it seems that theoretical regret bounds have been elusive for SGD-based approaches. In either case, the proof technique makes conceptual sense.

I have some clarifying questions and comments about the specifics:
1. Is the dependence on T the right dependence, meaning it matches the respective dependence without privacy, across all settings in Table 1?
2. What is the intuition behind the covariance offset in equation (4) scaling as sqrt(t)?
3. Can you clarify this point - “Although multi-parameter and single-parameter settings can be shown to be equivalent, they need independent analysis”? Why can one analysis not be translated into the other setting if they are equivalent?
4. Do you know if the dependence on d in the theorems is (sub)optimal?
5. Sometimes when you use $O$ you actually mean $\tilde O$. Please use these symbols consistently.
6. In line 127, please clarify which norm the context is bounded in (I guess $\ell_2$).
7. In Theorem 3.3, you use C1 and C2 in the text, but in the equation lines you use C in both cases.
8. In line 115, you set the problem up as having subgaussian noise, but then in line 127 you say that you can generalize to subgaussian and that the noise is bounded. Please make the problem setup consistent with these latter assumptions.
9. The notation paragraph defines a lot of standard concepts (such as the asymptotic notation). I think you could make it significantly shorter (and maybe save the extra space to add more details elsewhere).
10. There are some problems with the statement of Assumption 3. You define a* but don’t use it (note also the way it’s stated a* is a function of u), you also use v in the probability statement but it’s not defined.
11. There is something strange about the formulation of the single-parameter problem in line 123, as the outer product of e_i and \theta* is a matrix. The formulation in line 124 is clear.
12. In lines 26/27, you say “jointly DP”/“locally DP”. I believe the standard terms are “joint DP”/“local DP”.
13. In Theorem 3.2, you define Reg_\pi notation for an algorithm \pi, but then you don’t use this subscript notation in the end, in line 188.

---------------------------------------------------------------------------------------------------------------------------------------------------

I thank the authors for their thorough response. I think it would be helpful to add many of these details in the final version for the sake of clarity.

**Time Spent Reviewing:**

6

---

> ### Author Response · Authors · 2021-08-10
> **Response to Reviewer SZ1X**
>
> Thank you for your detailed reading and valuable comments. Please find our response to the comments below:
>
>
> **Q1: Does the dependence on $T$ matches the respective dependence without privacy in our single and multi param settings?**
>
> A1: In Table 1, since we choose $\tilde{O}$ to outline the main trend, our results do match the respective dependence without privacy in all settings. Specifically,
> + In single param settings, existing non-privacy **worst-case bounds** (under similar assumption as ours) are $O(\sqrt{T\log T})$ ([Kannan2018], [Sivakumar2020] and [Han2020]) while our bound is of $O(\sqrt{T\log\log T})$ (Theorem 3.1).
> + In single param settings, few works in the literature consider the **margin-dependent bounds** as ours.
>  The only work we can find is [Han2020], which considers the case that $\beta = 1$.
>  Their bounds are $O(\log T)$ while our bound matches theirs up to a $\log\log T$ factor (Theorem 3.3).
> + In multi param setting, the margin-dependent bound is considered in [Bastani2020], and our result matches theirs up to a $\log(T)$ factor (Theorem 4.1).
>
> **Q2: What is the intuition behind the covariance offset in equation (4) scaling as sqrt(t)?**
>
> A2: As we discussed in lines 180-181, the intuition is to ensure that the private sample covariance is positive semidefinite with high probability. The factor is in the scaling of $\sqrt{t}$ is because at the time $t$, the operator norm of the cumulative privacy noise matrix is in the scaling of $\sqrt{t}$ with overwhelming probability.
>
> **Q3: Can you clarify this point - Although multi-parameter and single-parameter settings can be shown to be equivalent, they need independent analysis? Why can one analysis not be translated into the other settings if they are equivalent?**
>
> A3: They are equivalent since each of their action selecting procedure can be transformed to the other.
> More specifically,
> + Single param to Multi param: $X_t=(x_{t,1}, x_{t,2},\cdots, x_{t,K})$ and $\theta_i=e_i \otimes \theta$.
> + Multi param to Single param: $x_{t,i}=(\mathbf{0},\cdots, X_t,\cdots, \mathbf{0})\in \mathbb R^{Kd}$ (the $i$th block is $X_t$ while other blocks are zero vectors) and $\theta = (\theta_1, \theta_2,\cdots, \theta_K)\in \mathbb R^{Kd}$.
>
> Thus if an algorithm does not need any randomness assumption in the context, then the algorithm can be applied in both single and multi param settings. That is why we can compare our algorithm with the algorithm in [36] in multi param experiments while their algorithm is proposed in the single-param setting.
> However, when assuming the context is random and making randomness assumptions, we need to consider single and multi param cases separately:
> + When we formulate a problem as a single-parameter bandit, we usually assume there is a known feature map $\phi$ that maps user's features $X_t$ and possible actions $a_{j}$ to $x_{t,j}:=\phi(X_t,a_j), j=1,2,\cdots,K$.
>  Thus, it is natural to make assumption on each margin distribution $x_{t,j}$ without any constraint on the correlation between different $x_{t,i}$ and $x_{t,j}$.
> + On the other hand, when we formulate a problem as a multi param bandit, we do not assume the existence of the feature map.
>  So the interaction between feature $X_t$ and each possible action $a_{j}, j=1,2,\cdots,K$ is formulated as $g(X_t^T\theta(a_j))$, with $\theta(\cdot)$ an unknown map.
>  Thus, it is natural to make assumptions (e.g., diversity) on $X_t$ in this scenario, as in [Bastani2017], [Bastani2020], [Goldenshluger2013].
>
> However, the assumptions particular to each setting cannot be transformed to the other setting.
>
> **Q4: Do you know if the dependence on d in the theorems is (sub)optimal?**
>
> A4:
> + In the single param setting, there is a gap in the dependence on the dimension between our upper bound ($d$) and the non-private case ($\sqrt{d}$).
>  We tend to conjecture that the gap in our results cannot be eliminated.
>  As shown in [Duchi2018] that the lower bound for the estimation error of logistic regression under LDP constraints does incur an extra $\sqrt{d}$ factor compared with the non-private case (and this also holds for linear regression), we believe such extra $\sqrt{d}$ will affect the regret bound.
> + In the multi param setting, the bound in [Bayati2017] and ours are both $O((\sqrt{d}/\kappa_l)^{\beta+1})$.
>  We cannot find other works which prove a better dependence on $d$ in a similar setting with ours.
>
> **Q5: What is the main technical difference in the linear case between our paper and previous work, e.g. [29] and [36]?**
>
> A5: The privacy protection of our paper and theirs are both based on the Gaussian mechanism in the linear case.
> However, our establishment of the regret bounds is significantly different from theirs.
> In particular, both [29] and [36] use the confidence-region-based method, and the Gaussian mechanism would change the exploration region, which is essential for their regret results.
> Instead, we control the estimation error under the privacy noise and prove the associated regret bound on it.
> Our settings and proof technique are entirely different.
>
> Although the linear case is a special situation of the generalized linear bandits, it serves as a warm-up for our presentation of SGD.
> Moreover, the linear case can be treated as a representation of those problems with an explicit formula of the optimal solution of MLE, while SGD provides a simple but effective solution for those without explicit expression.
>
> **Q6: In line 115, you set the problem up as having subgaussian noise, but then in line 127, you say that you can generalize to subgaussian and that the noise is bounded. Please make the problem setup consistent with these latter assumptions.**
>
>
> A6: Thanks for your suggestion. For readability, we can clarify it in our further version. Additionally, we will add the following remark on how to extend the bounded case to subgaussian case for clarification:
> "To extend the results for bounded contexts and noise to  subgaussian case,  note that the 1-subgaussian vectors is bounded by $O( \log T)$ with probability at least  $1-T^{-2}$.
> In practical, if the user finds the context or reward beyond this bound, she could upload a private version of zero context vector and zero reward to the server to protect her information.
> Since there are only $\log T$ occurrence, our regret guarantee in bounded case can still be held."
>
> **Other typos**
>
> Thanks a lot for point out other typos in our writing. We will replace our statement of "jointly DP" and "locally DP" with "joint DP" and "local DP".
>
> As for $Reg_{\pi}$, line 188 should be "we have for any possible $\varepsilon$-LDP algorithm $\pi$,
> $\sup_{\theta^{\star}:\lVert \theta^{\star}\rVert_2\leq 1 }\mathbb E[\text{Reg}_{\pi}(T;\theta^{\star})] = \Omega(\sqrt{T}/\varepsilon)$
> ".
>
> We will perform a thorough check of the whole paper.
>
> **References:**
>
> [Kannan2018] Sampath Kannan et al. "A Smoothed Analysis of the Greedy Algorithm for the Linear Contextual Bandit Problem"
>
> [Sivakumar2020] Vidyashankar Sivakumar et al. "Structured Linear Contextual Bandits: A Sharp and Geometric Smoothed Analysis"
>
> [Bastani2017] Hamsa Bastani et al. "Mostly Exploration-Free Algorithms for Contextual Bandits"
>
> [Bastani2020] Hamsa Bastani et al. "Online Decision-Making with High-Dimensional Covariates"
>
> [Han2020] Yanjun Han et al. "Sequential Batch Learning in Finite-Action Linear Contextual Bandits"
>
> [Goldenshluger2013] Alexander Goldenshluger et al. "A Linear Response Bandit Problem"
>
> [Duchi2018] John C. Duchi et al.  "Minimax Optimal Procedures for Locally Private Estimation"

---

### Official Review · Reviewer_5SWK · 2021-07-12

**Rating:** 7
**Confidence:** 3

**Summary:**

This work considers local differential privacy in stochastic generalized linear bandits with both single and multiple parameters, and thus develops LDP algorithms for both settings and analyzed the same regret bounds as in privacy-free setups.

**Limitations And Societal Impact:**

This work provides a complete piece of story with well-grounded theoretical result and experiments. I cannot find limitations on this work.

**Main Review:**

Originality:
LDP framework for multi-parameter setting seems novel and sophisticatedly designed so that it creates synthetic data for unselected arm to protect privacy from the server, and eliminates some inferior arms to avoid corruption caused by the synthetic update.

Quality:
The regret bounds for LDP algorithms seem theoretically well-grounded, and corresponding assumptions such as diversity and margin are the standard up to my knowledge.  It provides a complete piece in that it points out a clear theoretical gap in existing result by proving the lower bound, and closes this gap by proposing LDP algorithms. The numerical experiments also provides fair comparison to existing algorithms such as LDP-UCB and LDP-GLOC.

Clarity:
This submission is clearly stated in two different settings (single and multi-parameter) and well organized along with theoretical result and then numerical experiments.

Significance:
LDP is a state-of-the-art approach among those which allows statistical computations and protects each user's privacy. This is because unlike standard DP no trust in a server is necessary as noise is added to user inputs locally. It clearly solves the open problem : “can we close the gap between existing privacy result and non-privacy results?” by providing several LDP algorithm and obtaining the same regret bounds as non-private setup.

**Time Spent Reviewing:**

4

---

> ### Author Response · Authors · 2021-08-10
> **Response to Reviewer 5SWK**
>
> Thanks a lot for your supportive review on the novelty of our LDP mechanism in multi param case and our theoretical results.

---

### Official Review · Reviewer_ydgj · 2021-07-17

**Rating:** 6
**Confidence:** 3

**Summary:**

This paper combines generalized linear bandits and local differential privacy. Empirical experiments demonstrate that new method is superb in real datasets.

**Limitations And Societal Impact:**

Yes.

**Main Review:**

This paper is well-written and very clear in its motivation. The DP part of this paper looks legit though I didn't go through the details of the proof.

I have one questions: Lemma 2.1 seems incomplete to Theorem 3.22 [17], which holds only for $\epsilon\leq 1$. But in your experiments I see $\epsilon=5$ which should be guaranteed by Gaussian mechanism. Did I miss anything?

Also Equation (3) has a typo: it should be inverse of $M_t$.

I am no expert in contextual bandit so I cannot comment on this part.

**Time Spent Reviewing:**

2

---

> ### Author Response · Authors · 2021-08-10
> **Response to Reviewer ydgj**
>
> Thanks for finding our paper well-written and experimental results strong. The points you raised are explained in the following.
>
> **Q1: Why  Lemma 2.1 only holds for $\varepsilon\le 1$ while in our experiment we test performance under $\varepsilon=5?$**
>
> A1: Thank you for pointing out the glitch.
> We already update our experiments by replacing the results under $\varepsilon=5$ with those under $\varepsilon=0.5$, while other parameters remain the same, to demonstrate the effect of privacy constraints on the performance.
> Our LDP-OLS and LDP-SGD algorithms can still achieve superior performance under the stringent privacy constraint $\varepsilon=0.5$.
> We will update our paper according to your advice.
>
> |  | Single param |  Multi param  |  Real data |
> | :-----| :----: | :----: | :----: |
> | LDP-UCB | 269091.8 (256875.2, 281308.4) | 1171721.1 (1169951.6, 1173490.7)| N/A |
> | LDP-GLOC |  118252.3 (63563.8,172940.9) | 882358.4 (827849.9, 936866.9) | 67135.2 (67099.9, 67170.5) |
> | LDP-OLS | 82255.6 (50602.2,113908.9) | 373589.3 (361993.8, 385184.8) | N/A  |
> | LDP-SGD | 18749.7 (2281.5,39780.9)  | 705999.5 (654051.0，757947.9) | 18152.7 (11416.3, 24889.1) |
>
> **Q2: Whether Equation (3) has a typo: it should be inverse of $M_t$?**
>
> A2: As for the $M_t$ in Eq(3), it should be $M_t$ instead of $(M_t)^{-1}$ because in the following operations (i.e. in Eq (4)) we will first sum up $M_1$ to $M_t$, make a shift and then take an inverse of it.
> So we should collect $M_t$ instead of $(M_t)^{-1}$.
>
> Thank you for your patience and kindness which are very essential for the improvement of our paper.

---

### Official Review · Reviewer_9jC7 · 2021-07-17

**Rating:** 7
**Confidence:** 3

**Summary:**

Authors design LDP algorithms for stochastic generalized linear bandits to achieve the same regret bound as in non-privacy settings. The main idea is to develop a stochastic gradient-based estimator and update mechanism to ensure LDP. They also develop an estimator and update mechanism based on Ordinary Least Square (OLS) for linear bandits. Finally, they conduct experiments with both simulation and real-world datasets to demonstrate the consistently superb performance of algorithms to ensure strong privacy protection.

**Limitations And Societal Impact:**

Yes

**Main Review:**

The method is new and a combination of well-known techniques. The related works are adequately cited and it is clear for the comparison.
The submission is technically sound, and claims are well supported. Methods are used appropriately and it’s a complete work. The authors are careful and honest about evaluating both the strengths and weaknesses of their work.
The submission is clearly written and well organized.
The results are important and does advance the state of art.
There maybe some typos. Such as Line 115, what is this sigma?


**Time Spent Reviewing:**

5

---

> ### Author Response · Authors · 2021-08-10
> **Response to Reviewer 9jC7**
>
> Thank you for your review and positive comments on our paper. Please find our response to the comments below:
>
> **Q1: What is the sigma in line 115?**
>
> A1: Here $\sigma(X_1,\dots,X_{t},\epsilon_1,\dots,\epsilon_{t-1})$ means the sigma-field generated by random variables $X_1,\dots,X_{t}$ and $\epsilon_1,\dots,\epsilon_{t-1}$.
> Thank you for reminding us to add this notation into the "Notation" part of our paper in the future version.

---

### Decision · Program_Chairs · 2021-09-28

**Decision:**

Accept (Poster)

**Comment:**

Thanks for the submission; the reviewers were all positive about this paper.

**Consistency Experiment:**

NeurIPS has a long history of experimentation. In 2014, NeurIPS ran an experiment in which 10% of submissions were reviewed by two independent committees to quantify the randomness in the review process. This year, we repeated a variant of this experiment to see how the quality of the review process has changed over time.  This paper was part of the experiment and was therefore assigned to two committees (consisting of reviewers, an Area Chair, and a Senior Area Chair) that reached independent decisions.  If both committees made the same recommendation, this recommendation was followed. If a single committee recommended acceptance, the paper was accepted (with the exception of a few cases in which the other committee identified what we considered a fatal flaw, e.g., an error in a key result).

Both committees reached the same decision: **Accept (Poster)**

The other committee assigned to the paper recommended **Accept (Poster)**.  You can find the other set of reviews, along with any follow up discussion with the authors here:
https://openreview.net/forum?id=BEVDmheFG0